# Five Inhibitory Receptors Display Distinct Vesicular Distributions in Murine T Cells

**DOI:** 10.3390/cells12212558

**Published:** 2023-10-31

**Authors:** Jiahe Lu, Alisa Veler, Boris Simonetti, Timsse Raj, Po Han Chou, Stephen J. Cross, Alexander M. Phillips, Xiongtao Ruan, Lan Huynh, Andrew W. Dowsey, Dingwei Ye, Robert F. Murphy, Paul Verkade, Peter J. Cullen, Christoph Wülfing

**Affiliations:** 1School of Cellular and Molecular Medicine, University of Bristol, Bristol BS8 1TD, UK; jiahelu@fudan.edu.cn (J.L.); alisa.veler@weizmann.ac.il (A.V.); timsse.raj@med.uni-muenchen.de (T.R.); markchou1018@gmail.com (P.H.C.); lan.huynh@bristol.ac.uk (L.H.); 2Department of Urology, Fudan University Shanghai Cancer Center, Fudan University, Shanghai 200032, China; dwyelie@163.com; 3School of Biochemistry, University of Bristol, Bristol BS8 1TD, UK; boris.simonetti@crl.com (B.S.); p.verkade@bristol.ac.uk (P.V.); pete.cullen@bristol.ac.uk (P.J.C.); 4Wolfson Bioimaging Facility, University of Bristol, Bristol BS8 1TD, UK; stephen.cross@bristol.ac.uk; 5Department of Electrical Engineering & Electronics and Computational Biology Facility, University of Liverpool, Liverpool L69 7ZX, UK; a.m.phillips@liverpool.ac.uk; 6Computational Biology Department, School of Computer Science, Carnegie Mellon University, Pittsburgh, PA 15213, USA; xruan@berkeley.edu (X.R.); murphy@cmu.edu (R.F.M.); 7Bristol Veterinary School, University of Bristol, Bristol BS40 5DU, UK; andrew.dowsey@bristol.ac.uk; 8Shanghai Genitourinary Cancer Institute, Shanghai 200032, China; 9Department of Biological Sciences, Biomedical Engineering and Machine Learning, Carnegie Mellon University, Pittsburgh, PA 15213, USA

**Keywords:** inhibitory receptor, T cell activation, imaging, vesicular trafficking, proximity proteomics

## Abstract

T cells can express multiple inhibitory receptors. Upon induction of T cell exhaustion in response to a persistent antigen, prominently in the anti-tumor immune response, many are expressed simultaneously. Key inhibitory receptors are CTLA-4, PD-1, LAG3, TIM3, and TIGIT, as investigated here. These receptors are important as central therapeutic targets in cancer immunotherapy. Inhibitory receptors are not constitutively expressed on the cell surface, but substantial fractions reside in intracellular vesicular structures. It remains unresolved to which extent the subcellular localization of different inhibitory receptors is distinct. Using quantitative imaging of subcellular distributions and plasma membrane insertion as complemented by proximity proteomics and biochemical analysis of the association of the inhibitory receptors with trafficking adaptors, the subcellular distributions of the five inhibitory receptors were discrete. The distribution of CTLA-4 was most distinct, with preferential association with lysosomal-derived vesicles and the sorting nexin 1/2/5/6 transport machinery. With a lack of evidence for the existence of specific vesicle subtypes to explain divergent inhibitory receptor distributions, we suggest that such distributions are driven by divergent trafficking through an overlapping joint set of vesicular structures. This extensive characterization of the subcellular localization of five inhibitory receptors in relation to each other lays the foundation for the molecular investigation of their trafficking and its therapeutic exploitation.

## 1. Introduction

T cells become activated in the cellular interaction with antigen-presenting cells (APC). Recognition of a peptide/MHC complex on the APC surface by the T cell receptor (TCR) mediates immunological specificity and provides a strong stimulatory signal. This signal is tuned by the engagement of multiple costimulatory and coinhibitory receptors engaging ligands on the APC. An important challenge is to determine unique and overlapping functions among these coregulatory receptors. T cells express five key inhibitory receptors, CTLA-4, PD-1, LAG3, TIM3, and TIGIT [1,2,3]. While these receptors are also expressed in many other cell types, we focus on expression in T cells here. Inhibitory receptors can provide negative feedback in T cell activation as a default limitation of T cell activation, as is the case for CTLA-4 and PD-1 [4,5]. They can suppress T cell activation under defined physiological circumstances, for example, upon persistent antigen stimulation, as is the case for TIM3 and the highest levels of PD-1 expression [2]. A single T cell can express all five receptors at the same time. As the inhibitory receptors attenuate T cell activation, their signaling needs to be carefully controlled. A key control element is that a substantial fraction of the cellular pool of an inhibitory receptor is kept in vesicles in the cell interior, where the inhibitory receptors cannot engage their ligands on the APC surface. Regulated trafficking of vesicles containing inhibitory receptors to the cell surface in combination with endocytosis allows for control of their cell surface expression and, hence, function.

Vesicular localization of individual inhibitory receptors has already been characterized, that of CTLA-4 most extensively [6,7]. Briefly, more than 80% of the cellular pool of CTLA-4 is intracellular [8]. A cytoplasmic YVKM motif allows for constitutive AP-2-mediated endocytosis when not phosphorylated, limiting CTLA-4 cell surface expression in resting T cells and allowing for an increase upon tyrosine phosphorylation during T cell activation [6,8]. The association of endocytosed CTLA-4 with LPS-responsive beige-like anchor protein (LRBA) targets CTLA-4 for lysosomal degradation [9]. The association of CTLA-4 with TRIM, LAX, and Rab8 allows for regulated Golgi exit [10,11]. CTLA-4 can also be transported to the cell surface via secretory lysosomes [12] and be included in secretory microvesicles [13]. CTLA-4 trafficking is regulated by Rab5, Rab7 and Rab11 [14]. Upon T cell activation, the level of CTLA-4 at the cellular interface increases with a spatial preference for its center [15,16]. Ligand binding triggers CTLA-4 internalization to the extent that ligands can be stripped from the interacting APC [17]. A substantial part of PD-1 is located in Golgi-associated vesicles, and PD-1 accumulates at the cell surface upon T cell activation [18,19]. PD-1 can be shed via vesicles and rapidly endocytosed [20]. The majority of the cellular pool of LAG-3 is also located in vesicles overlapping with endosomal and lysosomal markers [21,22]. Upon T cell activation, trafficking is diverted from lysosomes to the cell surface [22]. Substantial intracellular localization of TIM3 has been described, including in Rab5 labeled vesicles and the *trans*-Golgi network, and TIM3 accumulates at the cellular interface with APCs [23,24]. However, substantial questions about inhibitory receptor localization remain unresolved. Are the subcellular distributions of different inhibitory receptors and the cellular and molecular determinants thereof distinct? Is inhibitory receptor insertion into the plasma membrane and their localization there coordinated? Here, we address these questions.

## 2. Materials and Methods

### 2.1. Mice

BALB/c CL4 TCR transgenic mice (RRID:IMSR_JAX:005307) [25] were bred and maintained at the University of Bristol under specific pathogen-free conditions. CL4 offspring were phenotyped by staining PBMCs with anti-CD8-APC (53-6.7, Biolegend, San Diego, CA, USA, RRID:AB_312750) and anti-Vβ8.2 FITC (KJ16-133, eBioscience, San Diego, CA, USA, RRID:AB_465261) antibodies. Homozygous B10/BR 5C.C7 TCR transgenic mice [26] (RRID:IMSR_EM:13156) were bred and maintained at the University of Bristol under specific pathogen-free conditions. BALB/c mice (RRID:MGI:2161072) were purchased from Yuxiu (Shanghai, China). All mice were culled for experimental use between 6 and 12 weeks of age.

### 2.2. Cells and Media

The complete medium consisted of RPMI-1640 plus L-glutamine (Gibco, New York, NY, USA) supplemented with 10% FBS (Brazilian source, ThermoFisher, Waltham, MA, USA), 50 µM β-Mercaptoethanol (Gibco), and PenStrep (Gibco) at 100 U/mL penicillin and 100 µg/mL streptomycin. Renca cells (RRID:CVCL_2174) were maintained in complete medium. Renca cells are negative for CD86 (unpublished) and CEACAM-1 [27] but express PD-L1 [28] and low amounts of Galectin9 [27]. The expression of ligands for LAG3 and TIGIT has not been determined. CL4 and 5C.C7 T cells were maintained in complete medium supplemented with 50 U/mL rh-IL-2 (NIH/NCI BRB preclinical repository) (‘IL-2 medium’). HEK293T (RRID:CVCL_0063) and Phoenix-E cells (RRID: CVCL_H717) were maintained in ‘Phoenix incomplete medium‘ consisting of DMEM with 4.5 g/L D-glucose, L-glutamine and sodium pyruvate (Gibco) supplemented with 10% FBS (Brazilian source, ThermoFisher), MEM non-essential amino acids (Gibco), and PenStrep (Gibco) at 100 U/mL penicillin and 100 µg/mL streptomycin. Long-term Phoenix cell stocks were kept in ‘Phoenix complete medium’, which is Phoenix incomplete medium supplemented with 300 µg/mL Hygromycin (Invitrogen, Waltham, MA, USA) and 1 µg/mL Diptheria toxin (Sigma, St. Louis, MO, USA).

### 2.3. Antibodies

Antibodies used are described in the order: antigen, fluorophore if applicable, clone, supplier, dilution, RRID:

For Western Blotting:

SNX1, polyclonal, Abcam, 1:1000, RRID:AB_296721

SNX2, 13, BD Biosciences, 1:1000, RRID:AB_398834

SNX5, polyclonal, Proteintech, 1:1000, RRID:AB_2192708

SNX6, D-5, Santa Cruz Biotechnology, 1:1000, RRID:AB_10842310

SNX27, polyclonal, Proteintech, 1:1000, RRID:AB_10888628

AP-1 γ adaptin, polyclonal, Proteintech, 1:1000, RRID:AB_2058209

AP-2 α adaptin, polyclonal, Proteintech, 1:1000, RRID:AB_2056316

VPS35, polyclonal, Abcam, 1:1000, RRID:AB_10696107

GLUT1, EPR3915, Abcam, 1:1000, RRID:AB_10903230

Polycystin-2, polyclonal, Santa Cruz Biotechnology, 1:1000, RRID:AB_2163376

Anti-mouse IgG, Alexa Fluor 680, Invitrogen, 1:20,000, RRID:AB_2633278

Anti-rabbit IgG, Alexa Fluor 800, Invitrogen, 1:20,000, RRID:AB_2633284

For immunostaining and T cell priming:

EAA-1, c45b10, Cell Signaling Technology, 1:200, RRID:AB_2096811

Anti-rabbit IgG, Alexa Fluor 568, Invitrogen, 1:400, RRID:AB_143157

CD3ε, 145-2C11, Biolegend, 5 µg/mL, RRID:AB_11149115

CD28, 37.51, BioLegend, 1 µg/mL, RRID:AB_11147170

CD3ε, FITC, 145-2C11, Biolegend, 1:50, RRID:AB_312671

CD4, PerCP/Cy5.5, RM4-5, BioLegend, 1:50, RRID:AB_893326

CD45, APC, 30-F11, BioLegend, 1:50, RRID:AB_312976

CD8α, APC/Fire750, 53-6.7, BioLegend, 1:50, RRID:AB_2572113

CTLA-4, PE, UC10-4B9, BioLegend, 1:50, RRID:AB_313254

PD-1, BV605, 29F.1A12, BioLegend, 1:50, RRID:AB_2562616

LAG3, APC, C9B7W, BioLegend, 1:50, RRID:AB_10639935

TIM3, PE/Cy7, RMT3-23, BioLegend, 1:50, RRID:AB_2571933

TIGIT, BV421, 1G9, BioLegend, 1:50, RRID:AB_2687311

### 2.4. Plasmids

For the retroviral expression of all receptors, stimulatory and inhibitory, we used the same previously described plasmid backbone where expression of a receptor with a C-terminally fused eGFP is driven by the Mouse Moloney Leukemia Virus long terminal repeat [29,30,31].

### 2.5. T Cell Isolation and Stimulation

Procedures used for the isolation and stimulation of TCR transgenic T cells have been extensively detailed elsewhere [28,29,30,31]. Briefly, for CL4 CD8^+^ T cells, red blood cells were removed from dissociated spleens using ACK lysis buffer (Gibco), and splenocytes were plated at a density of 5 × 10^6^ cells per well in 1 mL of complete medium in a 24-well plate. K^d^HA peptide (IYSTVASSL) was added to a final concentration of 1 μg/mL, and cells were incubated overnight. Primed cultures were stringently washed five times in PBS to remove unbound peptides and then retrovirally transduced if desired (detailed below). Subsequently, cells were re-plated at 4 × 10^6^ cells per well in 2 mL IL-2 medium in a 24-well plate. Thus, further CTL expansion was independent of TCR engagement. CTLs were sorted and imaged on day four. For 5C.C7 CD4^+^ T cells, dissociated lymph node cells were plated at a density of 5 × 10^6^ cells per well, in 1 mL of complete medium, in a 24-well plate. MCC peptide (ANERADLIAYLKQATK) was added to a final concentration of 3 µM. T cells were transduced after overnight culture (detailed below). Following transduction, cells were re-plated at 4 × 10^6^ cells per well in 2 mL IL-2 medium in a 24-well plate. T cells were sorted and imaged on day four or five.

For the flow cytometry experiments to quantify endogenous inhibitory receptor cell surface expression, a 24-well cell culture plate was coated with 5 μg/mL αCD3 and 1 μg/mL αCD28 in phosphate-buffered saline (PBS) at 4 °C overnight. Red blood cells were removed from dissociated BALB/c spleens using ACK lysis buffer (Gibco), and splenocytes were plated onto a 100 mm tissue culture dish for 2 h at 37 °C. Non-adherent cells were taken out and seeded onto the αCD3/αCD28-coated plate at 5 × 10^6^ cells per well. Thus, CTLs were expanded with continuous TCR engagement. At 48 h post T cell activation, cells were taken out from the coated plate and stained.

### 2.6. T Cell Retroviral Transduction

Procedures and plasmids used for retroviral transduction of TCR transgenic T cells have been extensively detailed elsewhere [29,30,31]. Briefly, transfection of Phoenix-E retroviral producer cells with a GFP-sensor expression plasmid was achieved using calcium phosphate precipitation. After transfection, cells were cultured for a further 48 h; then the retrovirus-containing medium was collected and used to transduce T cells. Then, 24 h after setting up a T cell culture, T cells were resuspended in 2 mL retrovirus-enriched supernatant from transfected Phoenix-E cells. Protamine sulfate was added to a final concentration of 8 μg/mL. Plates were centrifuged for 2 h at 200× *g*, at 32 °C. Following centrifugation, the supernatant was replaced with an IL-2 medium.

### 2.7. Spinning Disk Confocal Imaging

#### 2.7.1. Imaging of Cell Couples

Approximately 72 h after retroviral transduction, T cell cultures were resuspended in ‘imaging buffer’ (10% FBS in PBS with 1 mM CaCl_2_ and 0.5 mM MgCl_2_) plus DRAQ7. A BD Influx cell sorter (BD Bioscience) was used to isolate GFP-positive T cells [29]. As target cells for CL4 CTL, 1 × 10^6^ Renca cells were pulsed with K^d^HA peptide at a final concentration of 2 µg/mL for 1 h. These pulsed Renca cells will be referred to as Renca^+HA^ cells. Glass-bottomed 384-well optical imaging plates (Brooks Life Science Systems) were used for all imaging experiments. Imaging was done at 37 °C using a Perkin Elmer UltraVIEW ERS 6FE confocal system attached to a Leica DM I6000 inverted epifluorescence microscope and a Yokogawa CSU22 spinning disk. A 40× oil-immersion lens (NA = 1.25) was used for all imaging experiments unless otherwise stated. Prior to imaging, 50 µL of imaging buffer was added to the well, followed by 5 µL of T cells. T cells were allowed to settle for several minutes. Once a monolayer of T cells had formed, 4 µL of Renca^+HA^ cells were carefully added to the top of the well. Images were acquired for 15 min. Every 20 s, a z-stack of 21 GFP images (1 µm z-spacing) was acquired as well as a single, mid-plane differential interference contrast (DIC) reference image. After imaging, GFP files were saved as individual z-stacks for each time point, and both DIC and GFP files were exported as TIFF files for subsequent analysis.

#### 2.7.2. Determination of Fluorescence at the Cell Edge Versus Interior and of Distribution Heterogeneity

Fluorescent signal distribution within cells was measured using the ModularImageAnalysis (MIA; version 1.1.1, required files are provided in the Appendix A) workflow automation plugin [32] for Fiji-ImageJ [33,34] software version 1.53q. The workflow first detected cells from DIC images using a custom model for the StarDist ImageJ plugin [35]. Detected cells were filtered, with only those falling within an accepted size range retained for further analysis. Each cell was then split in two, based on the distance to the edge of the cell, yielding “interior” and “cell edge” cell objects. For analysis, the image slice with the highest mean intensity was extracted from the acquired GFP image stack, passed through a Gaussian filter (sigma = 1 px), and subject to rolling ball background subtraction (radius = 100 px). For each cell interior and edge, the mean and integrated GFP intensity were measured. In addition to this, the intensity distribution within the entire cell was quantified using a greyscale implementation of Ripley’s K-function [36], with measurements taken in the range of 1–9 px at 1 px intervals. Ripley’s K-function compares the distribution of signal at different length scales to that which would be expected by a completely spatially random distribution of the same amount of signal. With this, it is possible to identify length scales at which the signal is either clustering or exhibiting exclusion. Finally, the signal was measured using the Gini coefficient, where the Gini coefficient is a statistical descriptor of the inequality of a population. Coefficient values are given in the range 0 to 1, with these values corresponding to the cases where all pixels have equal intensity and where one pixel has all the signal (and every other has zero signal), respectively.

##### Vesicular Reorientation towards Interface

Live cell imaging of CL4 T cell MTOC reorientation has been described before [28]. Briefly, DIC files and GFP z-stacks were imported into Metamorph image analysis software version 7.7.0 (Molecular Devices). For each imaging time point, the GFP z-stack was assembled into a maximum projection. CTLs forming conjugates with Renca^+HA^ target cells were identified, and the time point at which a tight-cell couple forms was assessed using the DIC reference images. Tight cell couple formation was defined as the first time point at which a maximally spread immune synapse forms or two frames following initial cell contact, whichever occurred first. T cells were divided into three equally wide sections from the cellular interface backward. If a single cluster of vesicles could be detected, the position of the center of the cluster was recorded as part of one of the three sections.

##### Imaging of Individual T Cells to Estimate Association of Inhibitory Receptors with the Cell Edge

CL4 CTL retrovirally transduced to express inhibitory receptor GFP fusion proteins were imaged on an Olympus IXplore SpinSR imaging system with a 60× oil immersion lens (NA = 1.50) at 37 °C. A z-stack of 65 GFP images (0.25 µm z-spacing) was acquired, as well as a single, mid-plane differential interference contrast (DIC) reference image. The association of inhibitory receptor-GFP fluorescence with the cell edge was measured using the ModularImageAnalysis (MIA; version 1.2.7, required files are provided in the Appendix A) modular workflow plugin for the Fiji-ImageJ [33,34,37] software version 1.53t. The raw image stack was intensity normalized; then cells were enhanced in each slice using a U-Net pixel classification model run via integration with the DeepImageJ plugin (version 2.1.16) [38,39] and a custom model. The resulting probability image was binarized at a probability of 0.5, and both foreground and background regions in the binary image smaller than 200 px^3^ were removed. Cell regions that had become merged in the binary image were then separated using a distance-based watershed transform [40]. Connected components labeling was applied to the binary image stack to create 3D instances of the cells [40], and any detected cells larger than 1800 μm^3^ were discarded as these were assumed to correspond to multiple merged cells that had failed to separate. Any cells in contact with the image edge were also removed from further analysis. Fluorescence measurements were taken on the central slice of each cell, where the membrane is closest to being perpendicular to the imaging plane. First, the segmentation of the central slice of each cell was refined. Multiple radial intensity profiles were extracted, and the refined cell surface was taken to be the point at which, when starting from outside the cell, the fluorescence intensity first reached 30%/50% (inhibitory receptors/TCRζ) of the maximum intensity along that profile. The membrane-to-interior ratio of fluorescence intensity for each refined cell slice was calculated by splitting the cells in two based on distance from the cell surface. For this, the width of the membrane region was estimated by fitting orthogonal Gaussian profiles around the refined cell surfaces. To estimate membrane width for each cell, the smallest 5/20 (inhibitory receptors/TCRζ) Gaussian sigma values were averaged and multiplied by 2/3 (inhibitory receptors but TIM3/TIM3, TCRζ). The fluorescence intensity within the membrane and interior regions was measured and extrapolated to three dimensions, assuming isotropy.

### 2.8. TIRF Imaging

To image vesicular insertion into the plasma membrane, T cells were activated by a flat glass surface coated with αCD3ε mAb and soluble ICAM-1. The glass bottom of a 384-well MatriPlate (Brooks Life Sciences, MA, USA) was washed with 50 μL 1% acid alcohol (5% (*v*/*v*)) 37% HCl in 100% ethanol) at room temperature for 15 min and baked at 60 °C for up to 30 min. Then, 50 μL of coating solution (10 μg/mL αCD3 (Clone 145-2C11, BioLegend) + 2.5 μg/mL ICAM1-Fc (Novus Biologicals, Centennial, CO, USA, #720-IC) was incubated overnight at 4 °C. Alternatively, the glass bottom of a 384-well MatriPlate was coated with 1 mg/mL biotin-BSA (Sigma-Aldrich, St. Louis, MO, USA), followed by 200 µg/mL NeutrAvidin (ThermoFisher) and then 10 µg/mL biotin hamster anti-mouse CD3εantibody (clone 145-2C11, BioLegend) and kept in the imaging buffer (1 mM calcium chloride, 500 µM magnesium chloride, 10% (*v*/*v*) FBS). No differences in TIRF imaging between the two coating methods were observed, and data were therefore pooled. Live TIRF images were acquired using a Leica AM TIRF microscopy multicolor system attached to a Leica DMI 6000 inverted epifluorescence microscope equipped with 488 nm and 561 nm laser lines. A Leica HCX PL APO 100× oil immersion lens (NA = 1.47) was used. Images were acquired at frame rate every 333 ms. For dual color experiments, the laser was switched between frames resulting in image acquisition in each color every 735 ms. The onset of T cell activation was determined as the first frame in a series of rapidly spreading TIRF signals.

*To determine the fluorescence distribution across the interface*, a line with a width of 20 pixels (4.6 µm) was drawn across the widest region of a cell and divided into ten segments. The mean fluorescence intensity of each segment was measured and normalized to the segment with the highest fluorescent intensity.

*To determine the frequency of vesicular insertion*, a machine learning-based image analysis algorithm was developed to automatically identify vesicular insertion events at the plasma membrane. The training data of more than 100 annotated events were chosen to show the rapid appearance of a fluorescence signal followed by a slower dispersion as a first category or the sustained maintenance of this signal as a second category. These changes in fluorescence intensity and dynamics are most consistent with the release of vesicular content and, less certain, the formation of a receptor cluster, respectively. Alternate explanations can’t be excluded. The algorithm contained four sequential steps. (i) Blur detection: During time-lapse TIRF imaging, the particle signal might become blurred due to focus drift. To resolve this issue, we developed a support vector machine (SVM) classifier [41] to identify and exclude those blurred frames. As a result, we only retained the sequences of unblurred images in the middle for subsequent analysis. In the classifier, we used a collection of focus measure functions (‘ACMO’, ‘BREN’, ‘CONT’, ‘CURV’, ‘DCTE’, ‘DCTR’, ‘GDER’, ‘GLVA’, ‘GLLV’, ‘GLVN’, ‘GRAE’, ‘GRAT’, ‘GRAS’, ‘HELM’, ‘HISE’, ‘HISR’, ‘LAPE’, ‘LAPM’, ‘LAPV’, ‘LAPD’, ‘SFIL’, ‘SFRQ’, ‘TENG’, ‘TENV’, ‘VOLA’, ‘WAVS’, ‘WAVV’, ‘WAVR’ from ‘https://www.mathworks.com/matlabcentral/fileexchange/27314-focus-measure’ (accessed on 1 September 2022) as the feature set. The training data was generated by human annotation of time series images. The parameters in the SVM were optimized using 10-fold cross-validation. (ii) Particle detection: The general idea of particle detection is to identify particles using a two-sample *t*-test for local maximum versus local minimum. First, the region of interest (ROI) for particle detection for all time points was determined by 90% of Otsu’s thresholding [42] on the averaged image of all time points (after removing the blurred frames). Then, for each frame, the image was smoothed using wavelet smoothing (Biorthogonal wavelet with 3 and 5 vanishing moments) [43]. After that, the candidate particles were determined as the local maximum within the ROI in the smoothed image; simultaneously, the local minimum points were identified from the smoothed image. Each candidate particle was checked as a valid particle or just noise via two-sample *t*-test. The sample for the candidate point was the pixel values of the five brightest pixels $x_{p}$ within the 3 × 3 patch centered by the candidate point. Then, the three closest local minimum points to the candidate point were selected, and the five dimmest points were extracted from each of the 3 × 3 patches centered on these local minimum points, forming the local minimum sample $x_{m}$. After that, the local background standard deviation $\sigma_{l}$ with the local maximum sample and global background standard deviation $\sigma_{g}$ with the lower 25% points in the whole frame were calculated. Then, the *p*-value of $x_{p}-4\sigma_{l}-3\sigma_{g}$ versus $x_{m}$ were calculated in the two-sample *t*-test. If the *p*-value is below a predetermined threshold (0.001 in this case), the point is considered a valid particle; otherwise, it is classified as a noise point. This process is repeated for all candidate points and all frames. (iii) Point tracking. After point detection, we used the simpletracker library (https://www.mathworks.com/matlabcentral/fileexchange/34040-simpletracker (accessed on 1 September 2022)) to track all the particles over time. We used the nearest neighbor algorithm in the linkage with a linkage threshold of $2\sqrt{2}$, and a gap threshold of 3 frames. (iv) Event classification. Tracked particles were classified into two categories. ‘Vesicle insertion’ events were characterized by rapid lateral dispersion of the fluorescence signal of a newly detected particle, ‘Microcluster’ events were stable. We don’t know what these events represent, possibly receptor microclusters. They were excluded from the subsequent counting of vesicle insertion events. After particle detection and tracking, we determined the event type with machine learning classification and prediction. The feature is the log10 of the *p*-value of each particle over the local background via a two-sample *t*-test. The training data was generated by human annotation of time series images. The SVM classifier was trained on the annotated data with 10-fold cross-validation. The events were predicted for each particle at each time point. The output file of this analysis was a list of events with onset time, duration, coordinates within the entire imaging field, and classification. As only a small fraction of the identified events consisted of rapid fluorescence appearance followed by sustained maintenance, all events were included in the analysis.

*For colocalization analysis in dual color TIRF experiments*, colocalization over time was measured using a workflow compiled in the ModularImageAnalysis (MIA, version 0.21.13, required files are provided in the Appendix A) workflow automation plugin [44] for Image J [33,34]. Cells in the two fluorescence channels were segmented starting with a 2D median filter (radius = 2 px) followed by rolling ball background subtraction (radius = 50 px). Filtered images were then binarized using local thresholding with the Sauvola algorithm (radius = 50 px) [45], and any holes or isolated fragments in the binarized images smaller than 25 μm^3^ were removed. Of the remaining regions in the binary images, only those with 2D circularity greater than 0.2 were retained. The binary images from the two input channels were merged and subject to an additional 2D median filter (radius = 2 px). Any remaining binary regions that had become merged were split apart using a distance-based watershed transform [40]. Individual cell objects were identified from the binarized images using connected components labeling [40], and any cells falling outside a 50 μm^3^ to 500 μm^3^ accepted size range were discarded. Cells were tracked through time [46], and any tracks detected in an insufficient number of frames were also discarded. Pearson’s Correlation Coefficient was measured within each cell object for the two raw fluorescence image channels.

### 2.9. APEX Proteomics and Electron Microscopy

*APEX proteomics:* 2 × 10^6^ CL4 T cells transduced and FACS sorted to express CTLA-4-GFP, LAG3-GFP, or TIM3-GFP were incubated with 500 μM biotin-phenol (Iris Biotech, Marktredwitz, Germany) in IL-2 medium at 37 °C for 30 min. Cells were collected and incubated with 1 mM H_2_O_2_ (Sigma-Aldrich) for exactly 30 s and spun for 30 s at 311 g. It should be noted that biotinylation can, in principle, occur both in the vesicular lumen and at the cell surface. However, as the majority of the cellular pool of the inhibitory receptors is intracellular and as extracellular labeling requires the use of the enzymatically more active unmodified horseradish peroxidase [47], the contribution of extracellular biotinylation is likely negligible. Cells were immediately washed four times with 200 μL quenching buffer (10 mM Sodium Azide, 10 mM Sodium Ascorbate, and 5 mM Trolox in 0.1 M PBS), once with 0.1 M PBS and left on ice. Cell pellets were then lysed at 4 °C with 80 μL lysis buffer (50 mM Tris, 0.5% (*v*/*v*) NP40, 1% (*v*/*v*) HALT^TM^ protease and phosphatase inhibitor cocktail (ThermoFisher), pH = 7.5) for 20 min. To the 20 μL lysis buffer, prewashed streptavidin beads (GE Healthcare, Chicago, IL, USA) were added per sample and rotated for 2 h. The beads were washed twice with wash buffer (50 Mm Tris, 0.25% NP40, 1% Protease + Phosphatase inhibitor cocktail) and then wash buffer without NP40 and processed for proteomic analysis. Biotinylated proteins were digested into peptides and labeled with TMT 6-plex tandem mass tags. Using liquid chromatography–tandem mass spectrometry, peptide sequence information was obtained by an MS1 scan, and then the reporter group of the TMT 6-plex labels was cleaved to detect their specific *m*/*z* by MS/MS whose intensity represents the relative amount of the peptide from the corresponding samples [48]. Mass spectrometry data were analyzed with seaMass, a protein-level quantification and differential expression analysis method [49]. The seaMass method generates protein-level quantifications for each sample through Bayesian mixed-effects modeling [50], accounting for variation due to digestion and sample-level variation and automatically infers normalization effects across and within iTraq/TMT plexes. To account for the correlation between paired positive and negative samples, an additional random effect was included in the experimental design. Differential expression estimates of the log-2-fold change of protein expression between each of the CTLA4, TIM3, and LAG3 conditions versus the negative control condition were generated. Seamass’s inferred per-sample protein-level quantifications were then used to generate t-sne plots [51], which show the similarity of the protein-level quantifications between samples.

*APEX electron microscopy*: 3 × 10^6^ Renca cells were pulsed with 2 μg/mL K^d^HA peptide for 1 h. At least 1 × 10^6^ APEX2-CTLA-4/LAG-3/TIM-3- GFP overexpressing CL4 T cells were sorted by FACS. Both cell types were suspended in 100 μL 10% FBS/PBS/1 mM CaCl_2_/0.5 mM MgCl_2_, mixed, and spun 30 s at 311 g. The supernatant was removed, and the cell pellet was incubated at 37 °C for 5 min. The cell pellet was fixed in 2.5% glutaraldehyde/3 mM CaCl_2_/0.1 M sodium cacodylate, pH 7.4 for 5 min at room temperature, then 1 h on ice. The cell pellet was washed with 0.1 M sodium cacodylate, pH7.4 (wash buffer), blocked in 30 mM glycine in wash buffer, then washed again in wash buffer, each step for 10 min. The wash buffer was replaced with a DAB + 0.03% H_2_O_2_ solution for 5 min. Cells were washed 10 min in wash buffer, followed by 30 min incubation with 1% osmium tetroxide. The cell pellet was washed with distilled water, fixed in 3% uranyl acetate for 20 min, washed with distilled water, and dehydrated through 70%, 80%, 90%, and 96%, and 3 times 100% ethanol, 10 min each. The pellet was infiltrated in 100% EPON resin, rotating overnight at room temperature, then embedded in fresh 100% EPON resin, polymerized in 12–24 h at 65 °C. Samples were cut into 20 μm sections and imaged by transmission electron microscopy.

### 2.10. Immunostaining

*Indirect immunostaining for EAA1*: 20,000 Renca cells in 50 µL complete medium were seeded in a 384-well plate (Corning) and incubated overnight at 37 °C. Renca cells were incubated with 1 µg/µL of HA peptide for 1 h. Then, 100,000 CL4 T cells transduced with GFP-tagged co-inhibitory receptors (LAG-3, TIM-3, CTLA-4, or PD-1) were added and left for 10 min at 37 °C. Cells were fixed in 50 µL 4% PFA for 20 min at 4 °C. Each well was washed with 50 µL of PBS. The reaction was quenched with 50 µL of 50 mM NH_4_Cl for 10 min at 4 °C, and wells were washed three times with 50 µL PBS. Cells were permeabilized using 50 µL of 0.02% Triton X-100 in PBS for 20 min at 4 °C. Cells were washed with 50 µL of PBS and incubated with 50 µL of blocking solution (1% BSA in PBS) for 30 min at room temperature. Cells were stained with 50 µL anti-EEA1 mAb (Cell Signaling Technology, Danvers, MA, USA) at a dilution of 1:200 in 1% BSA in PBS with Fc Block at the same dilution overnight at 4 °C. Cells were washed three times with 50 µL of PBS. The wells were incubated with 50 µL of secondary antibody (Anti-Rabbit IgG, Alexa Fluor 568) diluted 1:400 in 1% BSA in PBS with Fc Block at 1:200 for 1 h at room temperature. Wells were washed three times and left in 50 µL of PBS at room temperature. Nuclei were stained with Hoechst 33258 (Invitrogen) diluted 1:5000 in PBS for 10 min at room temperature.

*Lysotracker/Mitotracker staining*: 20,000 Renca cells in 50 µL complete medium were seeded in a 384-well plate (Corning) and incubated overnight at 37 °C. Renca cells were incubated with 1 µg/µL of HA peptide for 1 h. Then, 300,000 CL4 T cells transduced with GFP-tagged co-inhibitory receptors (LAG-3, TIM-3, CTLA-4, or PD-1) were suspended in IL-2 medium at 37 °C containing either 65 nM LysoTracker^®^ Deep Red (ThermoFisher Scientific) or 25 nM MitoTracker™ Red CMXRos for 1 h or 30 min, respectively. Following incubation, cells were resuspended in a 37 °C IL-2 medium. Then, 100,000 CL4 T cells were added and left for 10 min at 37 °C. Cells were fixed and stained with Hoechst 33258 as described in the EEA1 staining protocol.

*Image acquisition*: All images were collected with a Leica SP8 AOBS confocal laser scanning microscope attached to a Leica DMi8 inverted epifluorescence microscope. An HC PL APO CS2 40× oil immersion objective (NA = 1.3) was used. GFP constructs were excited with a 488 nm line of the white light laser, Hoechst-stained nuclei with the 406 nm line of the diode laser, and LysoTracker/EEA1/MitoTracker with the 561 nm line of the white light laser. The pinhole was set to 1 Airy unit (AU). A zoom factor of 6 was used to acquire all the images.

*Colocalization analysis*: All images were analyzed using the Fiji-ImageJ [33,34] software version 1.53q to determine the Pearson’s correlation coefficient (PCC) for overlap between EEA1/LysoTracker/Mitotracker and GFP-tagged co-inhibitory receptors.

Sets of images without Hoechst stain were analyzed using an image analysis workflow created using the MIA plugin (v0.18.2, required files are provided in the Appendix A) for the Fiji-ImageJ software. To determine the best focus slice from the z-stack, the GFP channel was filtered with a Difference of Gaussian (DoG) filter (radius = 6 px) to enhance the bright band around the edge of the cell. The Fiji Ridge Detection plugin was used to detect ridge-like objects [28,52]. The best focus slice was identified as the slice with the most “ridge” features—the longest summed ridge length. To identify membranes in the images, images were converted to RGB format. Pixel classification was applied using the WEKA Trainable Segmentation plugin to enhance membrane structures [53]. A 2D median filter (radius = 2 px) was applied to the membrane probability image based on the GFP signal. The image was binarized using a global Otsu threshold. Foreground and background regions smaller than 300 px^3^ in volume were removed [40]. A 2D erode binary operation (radius = 2 px) was applied to shrink objects. Membranes were detected as foreground-labeled pixels [40]. One analyzable object from each image was detected (including all cells qualifying for co-localization analysis). The PCC was calculated for detected membrane regions as detailed below.

Sets of images with Hoechst stain were analyzed using an image analysis workflow created using the MIA plugin (v0.18.15, required files are provided in the Appendix A) [54] for Fiji-ImageJ software [33,34] version 1.53q. To detect nuclei, a 3D median (radius = 2 px) was used to smooth the Hoechst 33258 image whilst retaining sharp edges on the nuclei. A 2D sliding paraboloid background subtraction (radius = 30 px) was performed on all z slices. The image was binarized using the global Huang threshold [55], followed by 2D hole filling and 2D median filtering (radius = 5 px). Nuclei were initially identified as contiguous regions of foreground-labeled pixels using connected components labeling [40]. Any objects detected with volumes less than 20 µm^3^ are excluded. Initial objects were updated to correspond to all pixels enclosed with an alpha shape (calculated using the MATLAB implementation) [56]. Adjacent nuclei objects that have become merged during binarization were separated using a distance-based watershed transformation [40]. To calculate the PCC for detected single-cell membrane regions, nuclei were removed from the foreground of the binarized cell image, leaving just the cytoplasm. The binarized cell image was sub-divided such that all points of a contiguous region were closest to the same nucleus. Using connected components labeling, cytoplasm objects were detected in the sub-divided, binarized images [40]. Red and green-channel intensities were measured for all points of the detected cytoplasm objects, and any objects with mean green-channel intensity lower than a user-defined threshold were excluded from further analysis. PCC was calculated for each object using the red and green channels.

*Immunostaining to determine the fraction of an inhibitory receptor on the cell surface*: 3 × 10^6^ day 3 BALB/c T cells were stained for surface receptors in 1% BSA/PBS for 30 min at room temperature. The surface-stained cells were washed twice with PBS. One third of the cells was directly analyzed by flow cytometry; two thirds of the cells were fixed with 750 μL fixation buffer (BioLegend, #420801) for 30 min, washed twice with 1× permeabilization buffer (10% (*v*/*v*) in ddH_2_O, BioLegend, #421002), stained again in 1× permeabilization buffer for 30 min at room temperature and analyzed by flow cytometry.

### 2.11. Immunoprecipitation

Based on the pEGFP.C1 vector (Clontech), plasmids for the CMV promoter-driven expression of fusion proteins of GFP with either only the inhibitory receptor cytoplasmic domain or the joint transmembrane plus cytoplasmic domain were generated. HEK293T cells were grown to 90% confluence in 15 cm culture dishes and transfected with 15 μg DNA of GFP-inhibitory receptor cytoplasmic domain or transmembrane plus cytoplasmic domain constructs using 1 μL of a 10 mM polyethyleneimine stock solution in 5 mL Opti-MEM I reduced Serum Medium (Life Technologies, Carlsbad, CA, USA, #31985). GFP-trap beads (Chromotek, Planegg, Germany, GTA20) were aliquoted to 1.5 μL per sample and equilibrated by three washes in IP lysis buffer (50 mM Tris/HCl, 0.5% NP40, complete protease inhibitor cocktail (ThermoFisher, #A32955), pH = 7.4). HEK2393T cells were washed twice with ice-cold PBS, lysed by prechilled IP lysis buffer on ice, and centrifuged at 13,000× *g* for 10 min at 4 °C. Two percent of the cell lysate supernatant was kept as an input sample, and the rest of the supernatant was incubated with the equilibrated GFP-trap beads for 1 h at 4 °C on a rotator. GFP-trap beads were washed three times with IP lysis buffer and once with IP lysis buffer without NP-40. Proteins were eluted from the beads and denatured in 2× sample buffer (4% SDS in lysis buffer) at 95 °C for 10 min. Proteins were detected by Western blotting using a LI-COR Odyssey scanner.

## 3. Results

### 3.1. Inhibitory Receptors Are Enriched in the Cell Interior with Varying Degrees of Clustering

To determine the subcellular distributions of inhibitory receptors, we generated fusion proteins of CTLA-4, PD-1, LAG3, TIGIT, and TIM3 with GFP at their C-termini and expressed the chimeric proteins in primary T cells using retroviral transduction. Such fusion proteins have previously been validated [18,20,22,57,58,59]. To allow comparison to a large established data set, we first used CD4^+^ 5C.C7 T cell receptor (TCR) transgenic T cells [26]. We sorted transduced cells to about 600,000 GFP molecules per cell [31] and imaged them with spinning disk confocal microscopy (Figure 1A). We segmented T cells in the differential interference contrast (DIC) image, identified the midplane of a three-dimensional GFP fluorescence stack, and divided the cell into their edge, comprising 30–40% of the midplane area, and their interior, the remainder (Figure 1A). For all five inhibitory receptors as compared to three receptors with established expression on the cell surface, CD2, CD28, and CD6, a significantly (*p* < 0.01) smaller fraction of the total fluorescence was located at the cell edge as opposed to the cell interior (Figure 1B). These data suggest enhanced vesicular localization of all inhibitory receptors. By visual inspection, receptor distributions displayed a different degree of clustering (Figure 1A). Using the Gini coefficient for quantification, the distribution of CTLA-4 was significantly (*p* < 0.01) more clustered than that of any other receptors except for CD2 (Figure 1C). The LAG3 distribution, while less clustered than that of CTLA-4, was significantly (*p* < 0.01) more clustered than that of the other three inhibitory receptors, PD-1, TIGIT, and TIM3 (Figure 1C). Ripley’s K analysis of the same data as an alternate means to quantify clustering yielded the same trends (Appendix A).

Inhibitory receptors play a substantial role in the killing of tumor target cells by cytotoxic T cells (CTL). We use the interaction of CTL from CL4 TCR transgenic mice with Renca renal carcinoma cells as a model of tumor cell cytolysis [25]. We have repeated the localization analysis in these CD8^+^ CL4 TCR transgenic T cells by determining inhibitory receptor localization in the absence of target cells as a first step (Figure 2A). Similar to 5C.C7 CD4^+^ T cells, a significantly (*p* < 0.05) smaller fraction of the total fluorescence of all five inhibitory receptors as compared to that of the TCR was located at the cell edge as opposed to the cell interior (Figure 2B). Using the Gini coefficient, the distribution of CTLA-4 was again more significantly (*p* < 0.0001) clustered than that of the other four inhibitory receptors and that of the TCR (Figure 2C). The distribution of LAG3, while less clustered than that of CTLA-4, was significantly more clustered (*p* < 0.05) than that of the TCR, PD-1, TIGIT, and TIM3 (Figure 2B). The same trends were seen using Ripley’s k analysis (Appendix A).

**Figure 1 cells-12-02558-f001:**
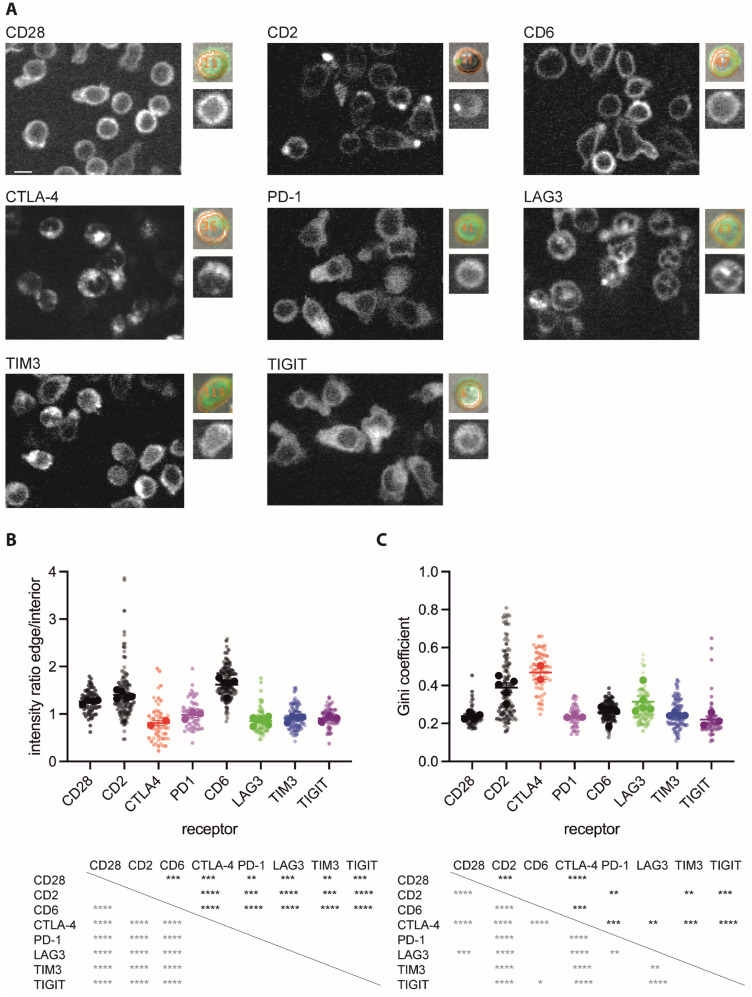
Inhibitory receptors are enriched in the interior of CD4^+^ T cells. (**A**) Representative spinning disk confocal midplane images of the GFP fluorescence of 5C.C7 T cells expressing a GFP fusion protein of the indicated receptor. The small images show, on the bottom, an individual cell and, on top, the matching overlay of the same GFP fluorescence in green, and the analysis masks for the cell edge and interior in orange over the corresponding DIC image. Scale bar = 5 µm. (**B**) The ratio of the midplane spinning disk confocal GFP fluorescence at the T cell edge over the interior of 5C.C7 T cells expressing a GFP fusion protein of the indicated receptor as mean ± SEM of run averages. Small symbols are individual cells with independent experiments denoted by color intensity. Large symbols are run averages. In total, 56 to 141 cells per receptor from 2 to 5 independent experiments. Statistical significance of differences between receptors is given in the table below, calculated based on run averages on top in black, based on individual cells on the left in grey. (**C**) Gini coefficients for the same T cells as in (**B**), given as in (**B**). * *p* < 0.05, ** *p* < 0.01, *** *p* < 0.001, **** *p* < 0.0001; *p* values calculated using one-way ANOVA.

To provide an estimate of which fraction of the inhibitory receptors is located at the plasma membrane versus the cell interior, we acquired a smaller image data set of live CL4 CTL expressing TCRζ or the five inhibitory receptors right at the resolution limit of confocal microscopy (Figure 3A). Even though the plasma membrane thickness is smaller than this resolution limit, these images allowed us to generate an automated image analysis routine approximating the fluorescence signal associated with the plasma membrane. Using this routine, only 7–28% of the cellular pool of the five inhibitory receptors was associated with the plasma membrane, significantly (*p* < 0.01) less than the 59 ± 2% of the TCR, a stimulatory receptor that nevertheless also continuously cycles through vesicular compartments [60,61] (Figure 3B). As an alternate approach, we activated T cells with beads coated with antibodies against CD3 and CD28 for 48 h and stained them at the cell surface and after permeabilization in the entire cell for the endogenous amounts of the five inhibitory receptors and CD45 as a plasma membrane marker (Figure 3C). While cell surface expression accounted for all of CD8 and the majority of the TCR, only 13–39% of the cellular pools of the inhibitory receptors CTLA-4, TIM3, and TIGIT were expressed at the cell surface (Figure 3D) in fairly close agreement with the image analysis data. In contrast, all of PD-1 and about half of LAG-3 were expressed at the cell surface. As PD-1 is moved to the cell surface upon lymphocyte activation [19,62], continuous TCR engagement in generating the activated T cells used for staining may have triggered PD-1 translocation to the plasma membrane here. In support, PD-1 is largely intracellular in the same T cells in the absence of TCR ligation (Appendix A). Acute TCR stimulation similarly triggers the translocation of LAG3 to the cell surface [22].

Upon binding of a T cell to an antigen-presenting cell (APC), the MTOC reorients within two minutes from behind the nucleus to the cellular interface [63,64]. To determine whether the distributions of the inhibitory receptors, as expected from vesicular localization, display similar relocalization, we activated CL4 CTL with Renca tumor target cells incubated with a high concentration, 2 µg/mL of the Influenza hemagglutinin peptide 518–526 and imaged the interaction with spinning disk confocal microscopy. A reorientation of vesicular clusters, to the extent that they could be detected, from behind the nucleus toward the cellular interface was consistently observed (Figure 3E,F). Together, these data are consistent with the suggestion that a substantial fraction of the cellular pool of all five inhibitory receptors resides in vesicles with distributions that combine some shared features with pronounced differences between different inhibitory receptors.

**Figure 3 cells-12-02558-f003:**
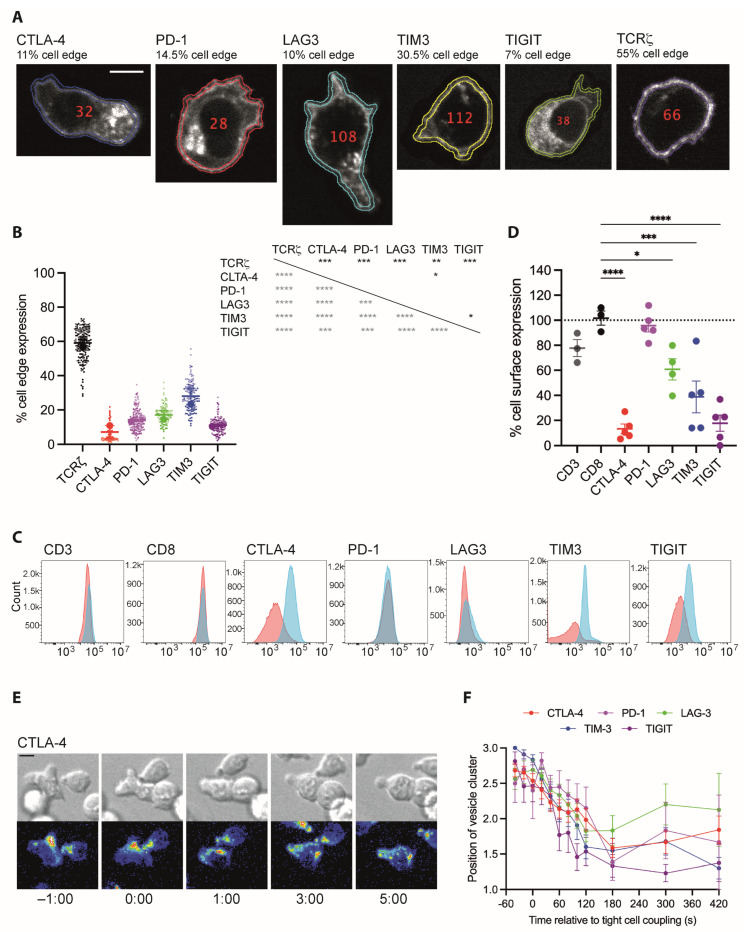
Inhibitory receptors are enriched in the interior of CD8^+^ T cells. (**A**) Representative spinning disk confocal midplane images of the GFP fluorescence of CL4 T cells expressing a GFP fusion protein of the indicated receptor. Scale bar = 5 µm. (**B**) The fraction of GFP fluorescence associated with the cell edge in CL4 T cells expressing a GFP fusion protein of the indicated receptor. Large symbols are averages of independent experiments with mean ± SEM and color intensity denoting matched experiments. Small symbols are the constituent single-cell measurements with independent experiments denoted by color intensity. A total of 77 to 131 cells per receptor from 2 independent experiments. Statistical significance of differences between receptors is given in the table to the right, calculated based on run averages on top in black based on individual cells on the left in grey. (**C**) Representative flow cytometry data of CD8^+^ T cells stimulated for 48 h stained for the indicated receptor on the cell surface (red) or in the entire cell (blue). (**D**) Fraction of endogenous inhibitory receptor expressed on the cell surface based on flow cytometry analysis (**C**) given as averages of independent experiments with mean ± SEM. In total, 3 to 5 independent experiments. (**E**) Representative time-lapse image sequence of a CL4 CTL expressing CTLA-4-GFP activating in the interaction with a Renca target cell incubated with 2 µg/mL HA peptide. DIC images on top, maximum projection images of the three-dimensional GFP fluorescence in a rainbow false color scale on the bottom, time given in minutes. Scale bar = 5 µm. (**F**) Average position of a single cluster of vesicles in cell couples of CL4 CTL expressing a GFP-fusion protein of the indicated receptor with a Renca target cell incubated with 2 µg/mL HA peptide over time as mean ± SEM. One is at the cellular interface, three is at the distal pole. In total, 15 to 41 cell couples per receptor from 3 independent experiments. * *p* < 0.05, ** *p* < 0.01, *** *p* < 0.001, **** *p* < 0.0001; *p* values calculated using one-way ANOVA.

### 3.2. Vesicles Transporting Inhibitory Receptors Are Inserted into the T Cell Plasma Membrane upon T Cell Activation with Distinct Frequency and Kinetics Resulting in Distinct Interface Distributions

As inhibitory receptors are localized in vesicles, one should be able to detect the insertion of such vesicles into the T cell plasma membrane upon T cell activation. To test this suggestion, we imaged the activation of CL4 T cells on glass coverslips coated with the stimulatory anti-TCR antibody 145-2C11 with or without the extracellular domain of the costimulatory ligand ICAM-1 using total internal reflection (TIRF) microscopy. CL4 T cells spread on the cover slips as an indication of activation (Figure 4A), and events consistent with vesicle insertion into the T cell plasma membrane, rapid appearance of fluorescence followed by lateral dispersion, were readily detectable (Figure 4B). We used a machine learning-based analysis routine to automatically detect such events during CL4 T cell activation. To capture dynamics, we tabulated insertion events in time windows of 100 frames, corresponding to about 30 s. The overall number of insertion events differed between inhibitory receptors, being the smallest for CTLA-4 and the highest for LAG3 (Figure 4C). Dynamics also differed. While the number of insertion events for CTLA-4 consistently increased over time, it consistently decreased for TIGIT (Figure 4D). For PD-1, LAG3, and TIM3, an initial burst of insertion events within the first 100 frames was followed by largely stable insertion thereafter (Figure 4D).

To determine whether the divergent insertion of vesicles carrying different inhibitory receptors resulted in distinct inhibitory receptor distributions across the T cell coverslip interface, we performed a line scan analysis of the TRIF data where we measured the fluorescence intensity in ten equally sized segments of a line of a width of 4.5 µm across the entire interface. The CTLA-4 distribution was focused consistently more on the center of the interface (Figure 5A–D). While LAG3 was slightly focused on the interface center at an early time point (first 100 frames of T cell spreading), it was uniquely enriched at the interface edge at the late time point (≥5 min after initial spreading) (Figure 5A–D). Distributions of PD-1, TIGIT, and TIM3 were similar, with a slight focus on the interface center that decreased over time (Figure 5A–D). To confirm that the divergent distributions derived from the single-color analysis of the five inhibitory receptors could be recapitulated in a two-color experiment, we imaged CL4 CTL expressing a fusion protein of LAG3 with tdTomato with either CTLA-4-GFP or TIM3-GFP (Figure 5E). Consistent with the selective focus of the CTLA-4 distribution on the interface center, the colocalization of LAG3 with CTLA-4 was less than that with TIM3 (Figure 5F). Consistent with the selective translocation of LAG-3 to the interface edge, colocalization between LAG3 and TIM3 declined over time (Figure 5F). The two selected two-color experiments thus validate the single-color analyses.

**Figure 4 cells-12-02558-f004:**
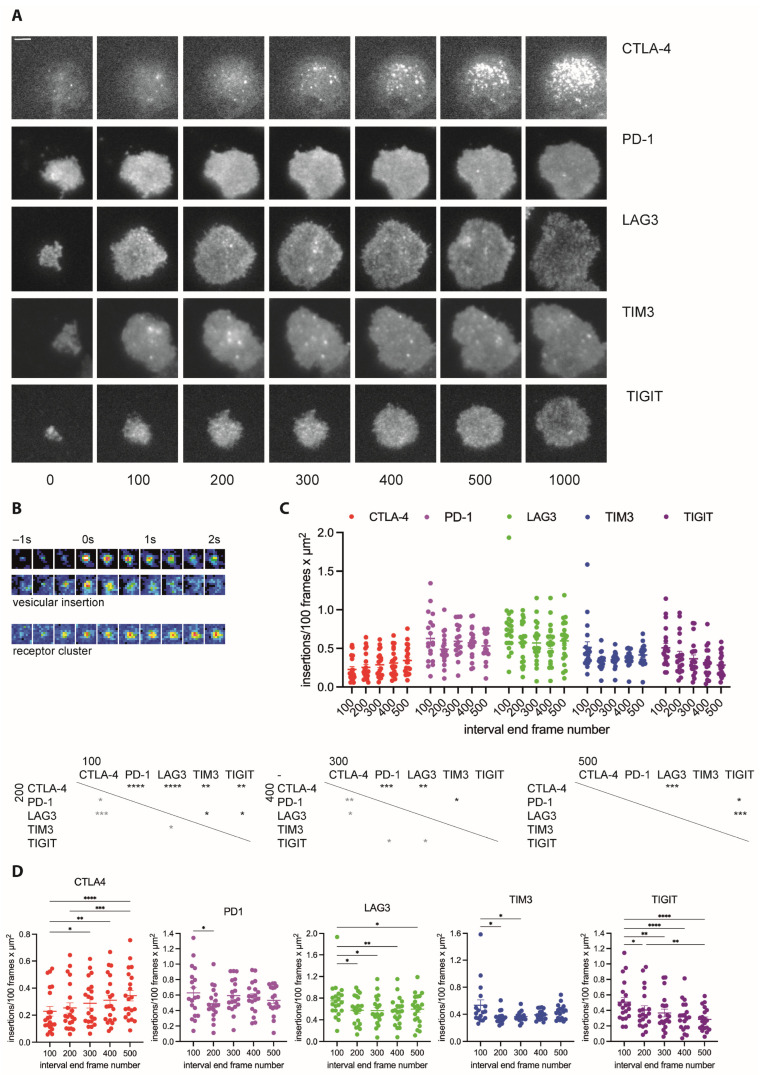
Inhibitory receptors are inserted into the plasma membrane during T cell activation. (**A**) Representative time-lapse TIRF images of the GFP fluorescence of CL4 CTL expressing a GFP fusion protein of the indicated receptor activated on glass coverslips coated with αCD3ε with (LAG3) or without ICAM-1 (rest of the data). Time given in seconds. Scale bar = 2 µm. (**B**) Time-lapse TIRF image sequence of the GFP fluorescence in a false color rainbow scale at frame rate of two membrane insertion events characterized by rapid appearance of the fluorescence signal followed by dispersion (**top**) in contrast to a stable receptor clustering event characterized by the more gradual appearance and a signal maintenance (**bottom**). (**C**) For the experiments shown in A, number of insertion events per 100 frames and µm^2^ for the indicated receptors as individual cells plus mean ± SEM. Time is denoted by the number of the last frame in the time interval. Here, 100 frames correspond to 30 s. Coverslips were coated with (part of the LAG3 data) or without ICAM-1 (rest of the data). All LAG3 data were pooled as detailed in (**D**). Here, 19 to 21 cells from 4 to 6 independent experiments. Statistical significance of differences between receptors is given in the tables below for each of the five time windows as indicated. (**D**) The same data as in C displayed in separate panels for the individual receptors with statistical differences between time windows. For LAG3, data from coverslips with or without ICAM-1 are indicated by light/dark shades of green, respectively. * *p* < 0.05, ** *p* < 0.01, *** *p* < 0.001, **** *p* < 0.0001; *p* values calculated using one-way (**D**) and two-way (**C**) ANOVA.

In summary, across all imaging assays, the five inhibitory receptors displayed distributions consistent with a predominant vesicular localization. This contrasted with the stronger enrichment of the TCR, the costimulatory receptors CD28 and CD2, and CD6 at the cell edge. When analyzed in conjunction with all imaging assays, distinct localization profiles emerged for all inhibitory receptors (Figure 5G,H). CTLA-4 stood out with the highest degree of clustering, the lowest rate of vesicular plasma membrane insertion, the only insertion rate increasing over time, and strong central interface localization. PD-1, in contrast, combined little clustering with a high, stable rate of vesicular insertion. LAG3 clustering was second highest to that of CTLA-4, and high-rate vesicular insertion led to a unique accumulation at the interface periphery. The distributions of TIGIT and TIM3 were most similar, with little clustering and intermediate rate vesicular insertion with a slight preference for the interface center. Yet, while the rate of vesicular insertion of TIGIT was steadily declining, that of TIM3 was slightly increasing after a rapid decline of a pronounced initial peak.

### 3.3. The Proteomes of Vesicles Harboring CTLA-4, LAG3, and TIM3 Are Distinct

To corroborate distinct subcellular localization of inhibitory receptors in molecular terms, we used APEX2 proteomics [65,66], a biotin-based proximity-labeling approach, to label the lumen of vesicles containing inhibitory receptors. We tagged all five inhibitory receptors with the engineered peroxidase APEX2 at the extracellular domain, the domain pointing into the vesicle lumen, and with GFP at the cytoplasmic domain. These proteins were retrovirally expressed in primary CL4 CTL, and the transduced CTL was sorted for expression of the tagged receptors. Only expression of the APEX2-tagged versions of CTLA-4, LAG3, and TIM3 were sufficiently high to allow determination of the luminal vesicular proteome. Using streptavidin labeled with Alexa Fluor 568 to detect proteins biotinylated by APEX2, we confirmed that APEX2 labeling was specific for structures enriched in the inhibitory receptors (Appendix A). Using an unbiased analysis method based directly on peptide counts in mass spectrometry, the vesicular proteomes of CTLA-4, LAG3, and TIM3 were reproducible and distinct (Figure 6A). Selective enrichment of CTLA-4 and LAG3 in the respective proteomes supports the specificity of the analysis (Figure 6B). Enrichment of the transferrin receptor (Tfrc), CD45 (Ptprc), and Transmembrane p24 trafficking protein 10 (Tmed10) in all three vesicular proteomes supports endocytic trafficking of all three receptors as an overlapping part of their vesicular localization (Figure 6B). Thus, these proteomics data corroborate the imaging data showing that inhibitory receptors reside in vesicular populations that differ between the receptors. The shared enrichment of some vesicular proteins suggests that the receptors traffic at least partially through the same vesicle types, possibly with a divergent degree of enrichment. The identification of proteins uniquely enriched for individual receptors allows for the possibility of unique vesicle types harboring these receptors. However, the detected luminal proteins didn’t further allow the identification of specific types of vesicles harboring the inhibitory receptors, likely because the transport machinery mediating vesicular identity is largely bound to the cytoplasmic, not the luminal side of the vesicle membranes.

Our extracellular fusion of inhibitory receptors to APEX2 allows for the labelling of vesicles harboring the receptors in electron micrographs [66]. We could detect vesicles labeled with electron-dense material (Figure 6C). These data are consistent with the vesicular localization of inhibitory receptors. However, in a blind analysis of the entire data, the specificity of electron-dense labeling was not sufficiently high to allow quantification and detailed characterization of vesicle types by their morphology.

### 3.4. CTLA-4 Associates More Extensively with Lysosomes

To further characterize vesicles harboring inhibitory receptors, we incubated CL4 CTL expressing GFP fusion proteins of the inhibitory receptors for 10 min with Renca target cells incubated with a high concentration, 2 µg/mL of HA agonist peptide. We then stained the cell couples with an antibody against EEA1 to label early endosomes and with Lysotracker and Mitotracker to label lysosomes and mitochondria, respectively. We automatically segmented the cytoplasm of individual T cells (Appendix A) and determined the Pearson’s correlation coefficient between inhibitory receptor and vesicular marker distributions (Figure 7). TIGIT-GFP expression was not high enough to allow an effective execution of such an analysis. Correlation between the inhibitory receptors and individual vesicular markers was moderately positive, consistent with a distribution of inhibitory receptors across multiple vesicle types. The association of CTLA-4 with lysosomes was significantly (*p* < 0.05) higher than that of the other three receptors investigated (Figure 7A,B). As secretory granules of CTL are lysosomally derived, these data suggest that CTLA-4 selectively traffics with secretory granules. The association of inhibitory receptors with endosomes was overall highest for the vesicular markers (Figure 7C,D), suggesting consistent internalization of inhibitory receptors from the plasma membrane. In an exploratory experiment, we also investigated endosomal association at a single 20 min time point of T cell activation (Appendix A). At this time, CTLA-4 was significantly (*p* < 0.05) more strongly associated with endosomes than LAG3 and TIM3, reaching an average Pearson’s correlation coefficient of 0.45 ± 0.1. While a further investigation of the time dependence of vesicular associations is beyond the scope of this manuscript, the data suggest that vesicular associations vary over time and that such variation differs between the inhibitory receptors. We also investigated the association of inhibitory receptors with mitochondria, largely as a negative control but also as the APEX proteomics found enrichment of mitochondrial proteins such as the phosphate carrier Slc25a3 in the lumen of vesicles carrying LAG3. Colocalization with mitochondria was minimal, as expected (Figure 7E,F). Overall, the colocalization data identified a preferential association of CTLA-4 with lysosomally derived vesicles with potentially enhanced endocytosis late in T cell activation. Substantial differences in the localization of PD-1, LAG3, and TIM3 were not detected, possibly because these receptors are distributed across multiple vesicle types.

### 3.5. Association with Regulators of Receptor Transport Is Largely Shared across the Five Inhibitory Receptors

The cytoplasmic domains of the inhibitory receptors contain multiple sorting motifs conserved between mice and humans that may mediate the selective binding of key regulators of receptor trafficking (Figure 8A). To determine the ability of the cytoplasmic domains of the inhibitory receptors to recruit regulators of receptor trafficking, we expressed GFP fused to the cytoplasmic domains of the inhibitory receptors in HEK-293T cells, precipitated the GFP fusion proteins and probed for associated regulators of receptor trafficking. A punctuate distribution of the inhibitory receptor cytoplasmic domain-GFP proteins suggested some vesicular association (Figure 8B). However, we could not selectively precipitate a number of key regulators of receptor trafficking with these fusion proteins (Figure 8C–E).

To strengthen the association with vesicular structures, we generated fusion proteins of GFP followed by the transmembrane and cytoplasmic domains of the inhibitory receptors and repeated the pull-down experiments. The distribution of these fusion proteins appeared more punctuate (Figure 8F), consistent with enhanced vesicular association. With these chimeras, we detected the association of the inhibitory receptors with AP-1, AP-2 (Figure 8H,I), the ESCPE-1 subunits SNX1/2/5/6 [67] (Figure 9A–E) and the core Retromer subunit VPS35 [68] (Figure 9F,G). The transmembrane/cytoplasmic domain constructs of all five receptors are efficiently associated with AP-1 and AP-2 (Figure 8H,I), suggesting that all receptors traffic through key conserved secretory and endocytic routes. Association with all sorting nexins and VPS35 was found for all inhibitory receptors, yet it was consistently highest for CTLA-4 without the differences between the inhibitory receptors reaching statistical significance (Figure 9). CTLA-4 association reached levels close to those of the cytoplasmic domain of the cation-independent mannose-6-phosphate receptors as a positive control for ESCPE-1-dependent retrograde transport. These data suggest that all inhibitory receptors engage with principal vesicular trafficking machinery. CTLA-4 engages with retrograde transport to a substantial extent, more so than the other inhibitory receptors.

**Figure 8 cells-12-02558-f008:**
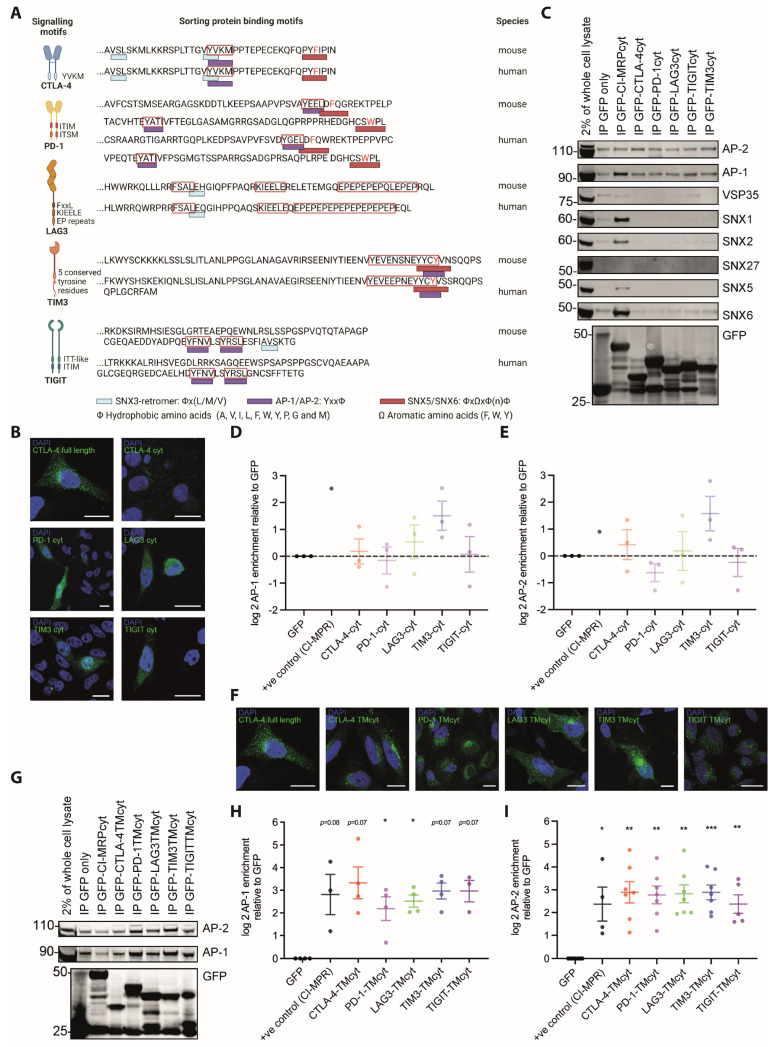
The cytoplasmic and transmembrane domains of the inhibitory receptors associate with AP-1 and AP-2. (**A**) Protein sorting motifs in the cytoplasmic domains of the inhibitory receptors. Figure panel created with BioRender.com. (**B**) Representative images of HEK293T cells expressing fusion proteins of the cytoplasmic domains of the inhibitory receptors and of full-length CTLA-4 for comparison (Hoechst 33258 in blue, GFP in green) as indicated. Scale bar = 10 µm. (**C**) Representative Western Blots of GFP pull-down experiments from the HEK293T cells in B as indicated on top, with antibodies against vesicular markers and GFP as a control for fusion protein expression as indicated on the right. Full-size blots in Appendix A. (**D**,**E**) Log2 of enrichment of AP-1 (**D**) and AP-2 (**E**) at the cytoplasmic domains of the indicated receptors fused to GFP relative to GFP only. Three independent experiments. (**F**) Representative images of HEK293T cells expressing fusion proteins of the cytoplasmic and transmembrane domains of the inhibitory receptors and of full-length CTLA-4 for comparison (Hoechst 33258 in blue, GFP in green) as indicated. Scale bar = 10 µm. (**G**) Representative Western Blot of GFP pull-down experiments from the HEK293T cells in F as indicated on top with antibodies against AP-1, AP-2, and GFP as a control for fusion protein expression as indicated on the right. Full-size blots in Appendix A. (**H**,**I**) Log2 of enrichment of AP-1 (**H**) and AP-2 (**I**) at the cytoplasmic and transmembrane domains of the indicated receptors fused to GFP relative to GFP only. Three to seven independent experiments. A fusion protein of GFP with the cytoplasmic domain of cation-independent mannose 6-phosphate receptor (CI-MPRcyt) is used as the positive control for AP-1 and AP-2 binding. * *p* < 0.05, ** *p* < 0.01, *** *p* < 0.001 relative to GFP only; *p* values calculated using one-way ANOVA.

## 4. Discussion

Here, we have shown that the majority of the cellular pool of five inhibitory receptors resides in the cell interior (Figure 3) in contrast to the four receptors described to be expressed on the cell surface. Comparing the inhibitory receptors to each other in subcellular clustering, the dynamics of plasma membrane insertion, spatiotemporal distribution on the cell surface, and molecular protein neighborhood, they were shown to have distinct subcellular distributions. These data are consistent with previous studies showing that different members of receptor families, such as IP_3_ receptors [69], or functionally related groups of receptors, such as ion channels in neurons [70], can show distinct distributions across vesicular structures that adapt to the cellular activation state. Nevertheless, our data on inhibitory receptor distributions in T cells raise important questions.

Our data are only the start of a comprehensive investigation of the subcellular localization of inhibitory receptors. Future work will have to characterize in detail how such distributions change during T cell activation as a function of time. The dynamic data (Figure 3E) and the data on colocalization with vesicular marker after 10 min of T cell activation (Figure 7) show that inhibitory receptor localization remains predominantly intracellular during T cell activation. However, any quantitative and qualitative changes in such distributions remain to be determined. Similarly, changes to inhibitory receptor distributions with changing T cell activation conditions remain unresolved. As the system subcellular organization of T cell signaling is exquisitely sensitive to T cell activation conditions [31,71,72], we expect the same to apply to inhibitory receptor distributions. Comparing inhibitory receptor distributions in CTL initially activated for 24 h with peptide/APC and then expanded with IL-2, i.e., in the absence of further TCR engagement (Figure 1 and Figure 2), with distributions in CTL stimulated 48 h continuously with αCD3 plus αCD28 (Figure 3) indicates that this may be the case. While predominant localization in the cell interior was maintained for most inhibitory receptors, PD-1 moved from the cell interior to the cell surface upon CTL generation with continuous TCR engagement. A substantial omission of our studies is that we don’t investigate inhibitory receptors in the activation of naïve T cells. CTLA-4 and PD-1 are rapidly upregulated upon activation of naïve T cells and control their expansion. The generation of new tumor-reactive T cells is a key feature of the mechanism of action of antibodies to block PD-1 [73]. Thus, future experiments will have to determine in detail how inhibitory receptor distributions vary with T cell activation states. Such studies should also extend to human cells. Nevertheless, even with a very limited investigation of inhibitory receptor distributions over time and as a function of T cell activation condition, our data, as presented, already raise important questions.

Which processes limit plasma membrane localization of inhibitory receptors? A likely means to limit plasma membrane expression is constitutive endocytosis. Consistent with this suggestion, the endosomal marker transferrin receptor was highly enriched in the vicinity proteomes of all three inhibitory receptors studied, CTLA-4, LAG3, and TIM3 (Figure 6B). Moreover, the endocytic adaptor AP-2 could be pulled down by the transmembrane plus cytoplasmic domains of all five inhibitory receptors (Figure 8I). The association of a non-phosphorylated YVKM motif in the cytoplasmic domain of CTLA-4 is likely to mediate such binding [6,8].

How can distinct vesicular distributions of the five inhibitory receptors be maintained? Conceptually, distinct vesicular distributions can be achieved through receptor-specific recruitment into defined subsets of vesicles or through differential association with the same set of vesicular structures. Given the highly interconnected, dynamic, and molecularly complex nature of the vesicular system, these two options may simply be two ends of a continuous spectrum. Even though marker proteins such as EEA-1 or LAMP-1 are commonly used to distinguish different vesicle types, there are extensive contacts between the different vesicle types and vesicles with microtubules as principal mediators of vesicular trafficking [74,75,76]. In biochemical fractionation of vesicle types, marker proteins are enriched in single fractions but commonly also appear across a wide range of other fractions [77]. Through consideration of an increasing number of vesicular properties, further vesicle types can be defined in the interconnected and dynamic system, up to 35 in recent imaging-based work [78]. Molecular diversity further blurs the distinction between specific vesicle types and differential distribution across a dynamic continuum of structures. Proteomics identified 1285 different proteins associated with vesicles in HEK293T cells [79]. An siRNA screen found 132 proteins with 247 interactions between them as strong regulators of endocytosis and 969 proteins with 55,070 interactions as less dominant regulators [80]. Nevertheless, vesicular subtypes with a distinct function can still be defined, such as a subset of endosomes marked by VSP18 that allows the retention of type I interferons for days [81].

It seems unlikely that distinct vesicle subtypes underpin the differential localization of the five inhibitory receptors in T cells. Sorting motifs in the inhibitory receptor cytoplasmic domains are overlapping (Figure 8A). Pulldown experiments consistently identified interactions between sorting adaptors and all five inhibitory receptors (Figure 8 and Figure 9), albeit with modest differences in the efficiency of association. Colocalization with vesicular markers again showed a consistent association of the markers with all four inhibitory receptors studied (Figure 7), again with modest differences in extent. Vesicular proteomes of the three inhibitory receptors studied were distinct yet with substantial overlap (Figure 6). Consistently differential association with common transport machinery and vesicle types may be sufficient to generate distinguishable subcellular distributions of the inhibitory receptors. Our data also suggest an additional mechanism to generate distinctions in localization. Cargo adaptors can be enriched in specific regions of the plasma membrane [82], and endocytosis can be regulated by cellular adhesion sites [83]. Given the distinct spatial distributions of some inhibitory receptors at the interface between T cells and activating coverslips (Figure 5), an uneven distribution of cargo adaptors and/or differences in the distributions of integrins mediating adhesion [84] may allow sorting of inhibitory receptors into different subsets of endosomes. Different endosome subtypes can then lead to differential downstream sorting [85,86]. Eventually, only a comprehensive characterization of the inhibitory receptor-associated proteomes combined with a functional investigation of the role of the associated proteins in vesicular trafficking will elucidate the mechanisms driving distinct subcellular localization of the five inhibitory receptors. A recent investigation of the TIM3 interactome found 37 high-confidence interactors [87]. Thus, a comprehensive elucidation of inhibitory receptor trafficking likely requires substantial effort.

Understanding the distinct distributions of the five inhibitory receptors has translational potential. Even without detailed mechanistic insight, it allows for unique roles of individual inhibitory receptors, even in T cells that express all five of them. Such unique roles are emerging. For example, in our work, blocking TIM3 but not PD-1 can reactivate suppressed CTL [27,28]. Further investigation of the vesicular trafficking of the inhibitory receptors may contribute to the therapeutic exploitation of such distinctions. In checkpoint blockade cancer immunotherapy, inhibitory receptors are targeted individually and in combination with blocking antibodies. Altering the trafficking of inhibitory receptors, e.g., slowing plasma membrane insertion or accelerating endocytosis, in the context of such therapies may be synergistic with receptor blockade. While processes shared between multiple cell types, such as basic vesicular trafficking, are difficult to target therapeutically, further investigation of the molecular mechanisms of inhibitory receptor transport may reveal T cell-specific elements that can be exploited therapeutically. As an example, engineering an anti-CTLA-4 antibody such that antibody-bound CTLA-4 is not routed from endosomes to lysosomes but allowed to recycle to the cell surface reduces immunotherapy-related adverse effects while maintaining therapeutic efficacy in mice [57].

## Figures and Tables

**Figure 2 cells-12-02558-f002:**
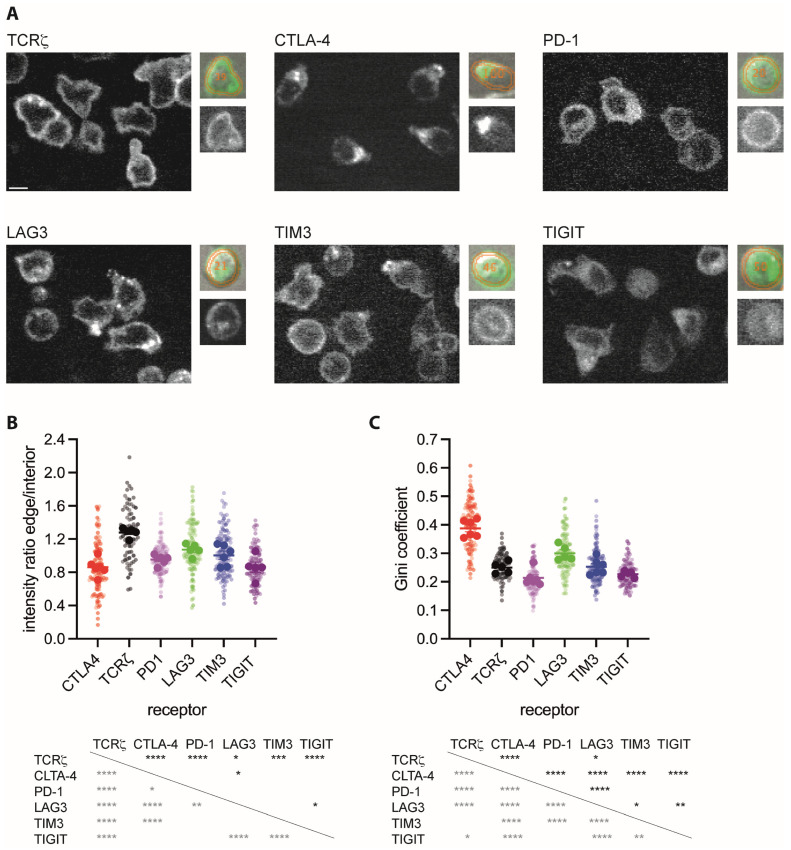
Inhibitory receptors are enriched in the interior of CD8^+^ T cells. (**A**) Representative spinning disk confocal midplane images of the GFP fluorescence of CL4 T cells expressing a GFP fusion protein of the indicated receptor. The small images show, on the bottom, an individual cell and, on top, the matching overlay of the same GFP fluorescence in green, and the analysis masks for the cell edge and interior in orange over the corresponding DIC image. Scale bar = 5 µm. (**B**) The ratio of the midplane spinning disk confocal GFP fluorescence at the T cell edge over the interior of CL4 T cells expressing a GFP fusion protein of the indicated receptor as mean ± SEM of run averages. Small symbols are individual cells with independent experiments denoted by color intensity. Large symbols are run averages. In total, 77 to 131 cells per receptor from 5 to 6 independent experiments. Statistical significance of differences between receptors is given in the table below, calculated based on run averages on top in black, based on individual cells on the left in grey. (**C**) Gini coefficients for the same T cells as in (**B**), given as in (**B**). * *p* < 0.05, ** *p* < 0.01, *** *p* < 0.001, **** *p* < 0.0001; *p* values calculated using one-way ANOVA.

**Figure 5 cells-12-02558-f005:**
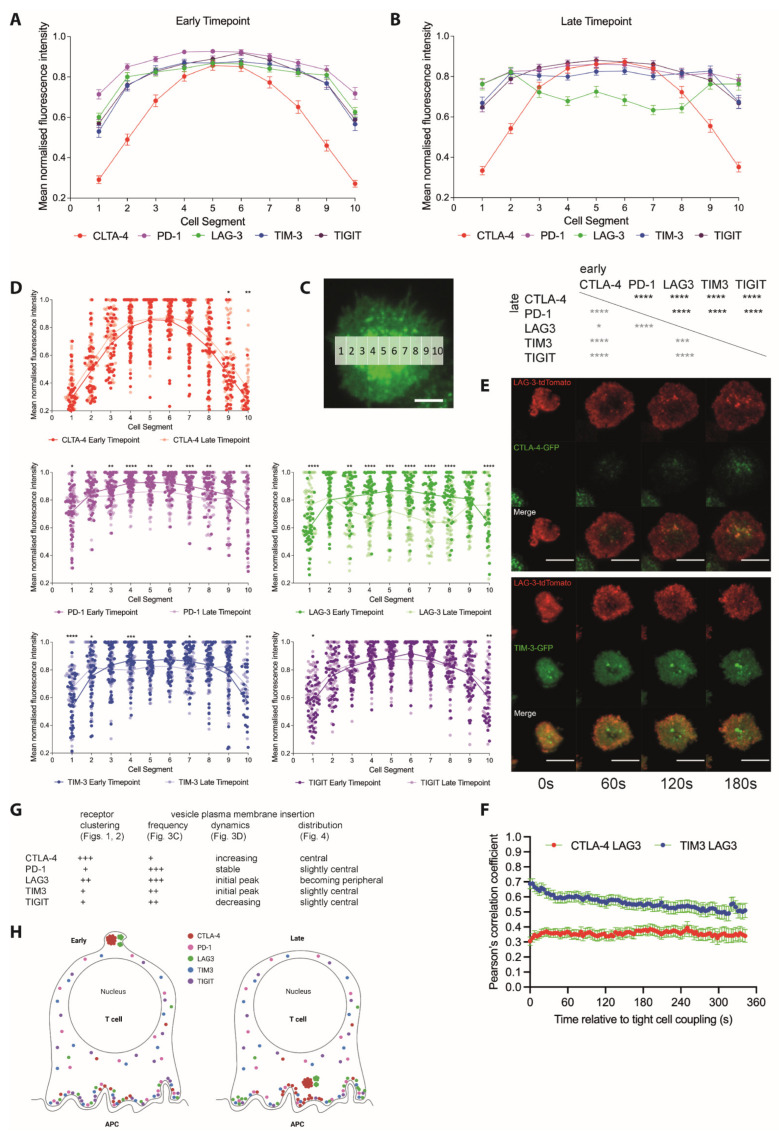
Inhibitory receptors are distinctly distributed in the plasma membrane during T cell activation. (**A**–**C**) CL4 CTL expressing a GFP fusion protein of the indicated receptor was activated on glass cover slips coated with αCD3ε and ICAM-1 and imaged by TIRF. Given is the GFP fluorescence intensity in ten equally sized segments of a line scan with a line width of 5 µm across the entire interface as mean ± SEM normalized as 1 for the segment with the highest intensity within 30 s (**A**) and >5 min (**B**) of CTL spreading. In total, 50 CTLs were analyzed per receptor from 4 to 6 independent experiments. Statistical significance of differences between receptors is given in the table separately for the early (in grey on the left) and late (in black on top) time points. (**C**) The representative image shows the 10 measurement regions. Scale bar = 5 µm. (**D**) The same data as in A are displayed separately for each inhibitory receptor with statistical significance of differences between early and late time points in each line scan segment. (**E**) Representative time-lapse TIRF images of the GFP and tdTomato fluorescence of CL4 CTL expressing the indicated fluorescent protein fusion protein of the indicated receptors activated on glass coverslips coated with αCD3ε and ICAM-1. Time given in seconds. Scale bar = 10 µm. (**F**) For the experiments shown in (**E**) and the indicated combinations of inhibitory receptors, Pearson’s correlation coefficients for GFP and tdTomato fluorescence across the entire interface as mean ± SEM over time. (**G**,**H**) Summary of imaging data from Figure 1, Figure 2, Figure 3, Figure 4 and Figure 5, as a table (**G**) and graphically as created with Biorender.com (accessed on 18 July 2023) (**H**). * *p* < 0.05, ** *p* < 0.01, *** *p* < 0.001, **** *p* < 0.0001; *p* values calculated using two-way ANOVA (**A**) and Wilcoxon matched-pairs rank test (**B**).

**Figure 6 cells-12-02558-f006:**
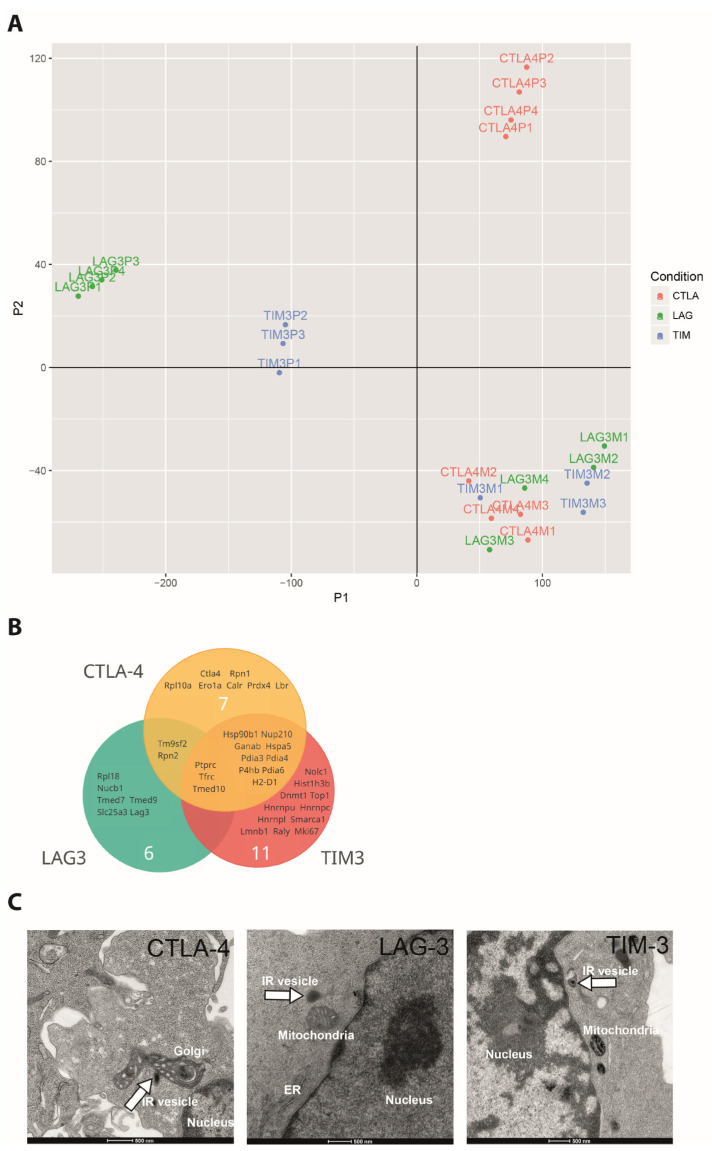
The vesicular proteomes of CTLA-4, LAG3, and TIM3 are distinct. (**A**) t-SNE plot of raw mass spectrometry peptide data after APEX2-mediated proximity biotinylation of proteins in the lumen of vesicles harboring APEX2 fusion proteins of CTLA-4, LAG3, and TIM3 expressed in CL4 CTL. Each dot is an independent experimental repeat labeled as P1 to P4 for the APEX2 experiments and as M1 to M4 for the no-APEX2 control for endogenously biotinylated proteins. (**B**) Gene names of proteins detected with a global false discovery rate of *p* < 0.05 in the lumen of vesicles expressing APEX2 fusions of CTLA-4, LAG3, and TIM3 and combinations thereof. (**C**) Examples of electron micrographs of CL4 CTL where vesicles expressing APEX2 fusion proteins of CTLA-4, LAG3, or TIM3 as indicated are labeled with an APEX2-mediated electron-dense precipitate as marked with a white arrow. Scale bar = 0.5 µm.

**Figure 7 cells-12-02558-f007:**
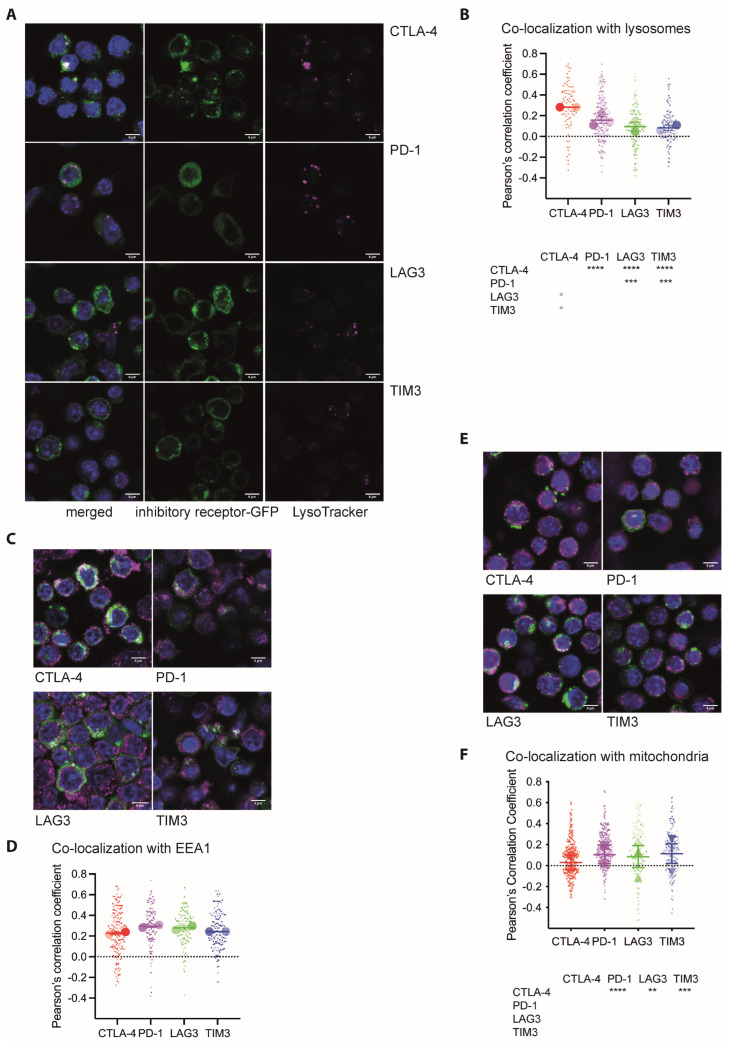
CTLA-4 is selectively enriched in lysosomally derived vesicles. (**A**) CL4 CTL expressing GFP fusion proteins of the indicated inhibitory receptors and incubated with LysoTracker Red DND-99 were activated by incubation with Renca target cells in the presence of 2 µg/mL HA agonist peptide for 10 min, fixed and stained for nuclei with Hoechst 33258. Given are confocal z-stack midplane images of the LysoTracker fluorescence (magenta), the GFP fluorescence (green), and merged images of all fluorescence data (Hoechst 33258 in blue). Scale bar = 6 µm. (**B**) For the experiment in A, Pearson’s correlation coefficients of the GFP and LysoTracker fluorescence distributions in the cytoplasm of individual CL4 CTL segmented from three-dimensional fluorescence data. Small symbols are individual cells with independent experiments denoted by color intensity. Large symbols are run averages. In total, 91 to 207 cells per receptor from 2 to 3 independent experiments. Statistical significance of differences between receptors is given in the table below, calculated based on individual cells on top in black, based on run averages on the left in grey. (**C**) CL4 CTL expressing GFP fusion proteins of the indicated inhibitory receptors were activated by incubation with Renca target cells in the presence of 2 µg/mL HA agonist peptide for 10 min, fixed and stained with αEAA1 and for nuclei with Hoechst 33258. Given are merged confocal z-stack midplane images of the αEAA1 (magenta), GFP (green), and Hoechst 33258 (blue) fluorescence. Scale bar = 6 µm. (**D**) For the experiment in C, Pearson’s correlation coefficients of the GFP and EAA1 fluorescence distributions in the cytoplasm of individual CL4 CTL segmented from three-dimensional fluorescence data. Small symbols are individual cells with independent experiments denoted by color intensity. Large symbols are run averages. In total, 101 to 166 cells per receptor from 2 independent experiments. There were no significant differences between the inhibitory receptors. (**E**) CL4 CTL expressing GFP fusion proteins of the indicated inhibitory receptors and incubated with MitoTracker Red CMXRos were activated by incubation with Renca target cells in the presence of 2 µg/mL HA agonist peptide for 10 min, fixed and stained for nuclei with Hoechst 33258. Given are merged confocal z-stack midplane images of the MitoTracker (magenta), GFP (green) and Hoechst 33285 (blue) fluorescence. Scale bar = 6 µm. (**F**) For the experiment in (**E**), Pearson’s correlation coefficients of the GFP and MitoTracker fluorescence distributions in the cytoplasm of individual CL4 CTL segmented from three-dimensional fluorescence data. Small symbols are individual cells with independent experiments denoted by color intensity. Large symbols are run averages. In total, 151 to 298 cells per receptor from 2 to 3 independent experiments. Statistical significance of differences between receptors is given in the table below, calculated based on individual cells on top in black, based on run averages on the left with no significant differences. * *p* < 0.05, ** *p* < 0.01, *** *p* < 0.001, **** *p* < 0.0001; *p* values calculated using one-way ANOVA.

**Figure 9 cells-12-02558-f009:**
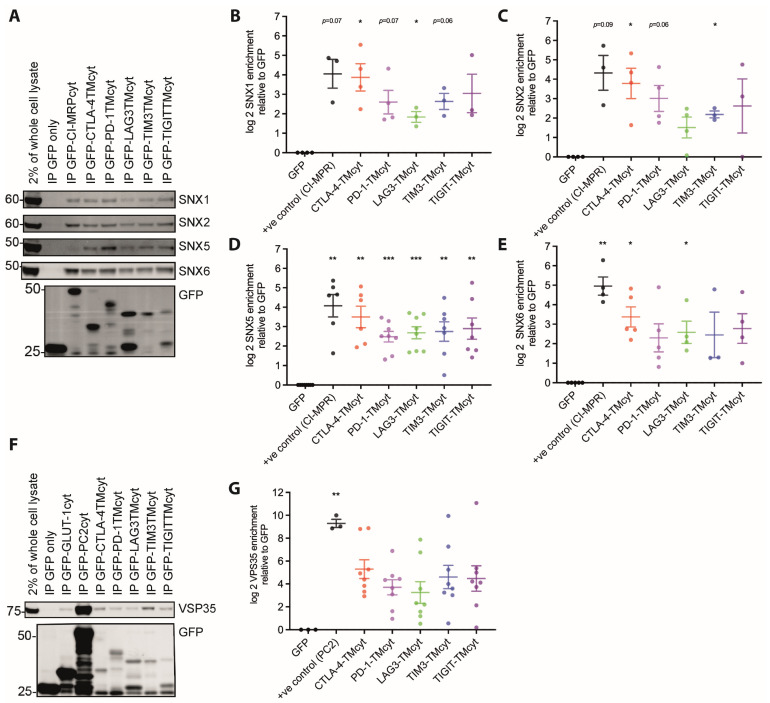
The cytoplasmic and transmembrane domains of inhibitory receptors associate with sorting nexins. (**A**) Representative Western Blot of GFP pull-down experiments from the HEK293T cells in Figure 8F as indicated on top with antibodies against SNX1/2/5/6 and GFP as a control for fusion protein expression as indicated on the right. Full size blots in Appendix A. (**B**–**E**) Log2 of enrichment of SNX1 (**B**), SNX2 (**C**), SNX5 (**D**), and SNX6 (**E**) at the cytoplasmic and transmembrane domains of the indicated receptors fused to GFP relative to GFP only. Three to eight independent experiments. (**F**) Representative Western Blot of GFP pull-down experiments from the HEK293T cells in Figure 8F as indicated on top with antibodies against VPS35 and GFP as a control for fusion protein expression as indicated on the right. (**G**) Log2 of enrichment of VPS35 at the cytoplasmic and transmembrane domains of the indicated receptors fused to GFP relative to GFP only. Eight independent experiments. A fusion protein of GFP with the cytoplasmic domain of cation-independent mannose 6-phosphate receptor (CI-MPRcyt) is used as the positive control for SNX-1 to SNX-6 binding. A fusion protein of GFP with the C-terminal cytoplasmic domain of polycystin-2 (PC2) is used as the positive control for VPS35 binding. * *p* < 0.05, ** *p* < 0.01, *** *p* < 0.001 relative to GFP only; *p* values calculated using one-way ANOVA.

## Data Availability

Raw imaging data are available at the University of Bristol data repository, data.bris, at https://doi.org/10.5523/bris.3ttc9epwoapu12o27ic0w8xxmz (accessed on 30 October 23); Raw proteomics data have been deposited to the ProteomeXchange Consortium via the PRIDE partner repository with the dataset identifier PXD044064.

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
