# Peer review of "Five Inhibitory Receptors Display Distinct Vesicular Distributions in Murine T Cells"

_cells, 2023, doi:10.3390/cells12212558_

Round 1

Reviewer 1 Report

Comments and Suggestions for Authors

This is an interesting work, well written and clear for the reader, describing the cellular distribution of the five inhibitory receptors on T cells.

I have only few comments:

- Could different stimuli and T cell differentiation/activation methods change the distribution of the five receptors or do you think that this localization is always the same?

- In the discussion session you mentioned only at the end the general impact that your findings could have on the use of checkpoint inhibitors. I would discuss a little bit more in detail this aspect, since it will empower the meaning of your results from a translational point of view.

Author Response

see attached file please

Reviewer 2 Report

Comments and Suggestions for Authors

This is interesting work which extends our understanding of regulation of inhibitory receptor expression on T cells. The authors confirm the well-established observation that CTLA-4 is mainly located to intracellular vesicles in effector T cells. Rather unexpectedly, they then suggest that a substantial pool of other T cell inhibitory receptors, such as PD-1, TIM3, LAG3, TIM3 and TIGIT, is also intracellular. This is important finding. However, there are several major problems with the data interpretation that should be addressed by the authors:

1) Over-expression of GFP-tagged receptors is used. How representative are the results of endogenous protein? The transduced T cells express endogenous inhibitory receptors (at least some of the 5 analysed), how does expression levels of these compare to the over-expressed GFP fusions?

2) Comparing surface stained vs permeabilized cells (Figure 3) – multiple antibodies show much higher background staining when used after permeabilization, as compared to surface staining. This could lead to overall higher signal in permeabilized cells for some antibodies. An important control is missing – comparison of surface and permeabilized (total) staining in antigen-negative cells.

3) Figure 4. How do the authors distinguish “insertion” of receptor into membrane from cell adhesion resulting in closer contact of the membrane to the glass, resulting in appearance of brighter GFP signal? Comparison the inhibitory receptor to a control protein with predominatly membrane localization would be an important control.

4) Proteome analysis (Figure 6). There is no specific enrichment step for vesicles, what is the evidence that the data shows vesicular proteome, rather than proteins in proximity of the inhibitory receptors at the cell membrane? Presence of the highly expressed membrane protein CD45 supports this alternative interpretation of the data.

Minor concerns/questions/comments:

1. The manuscript would benefit from some language editing. For example, “ T cells activate” is not correct, T cells become activated.

2. “PD-1 can be included into ectosomes and rapidly endocytosed” (L 73)

Incorporation of PD-1 into ectosomes does not lead to endocytosis; the referenced paper describes PD-1 endosome shedding and endocytosis under different experimental conditions

3. Are the TCR Tg mice on B6 or BALB/c background?

4.  Please mention/describe the plasmids used, even if they were used before (L 160)

5. Which specific protease + phosphatase inhibitor cocktail was used? What was the pH of lysis buffer? (L 343)

6. What is TCRz? (L 536)

7. Figure 2B. The text refers to CD8 + T cell interactions with Renca cell line, but Renca cells are not mentioned in the figure legend. Does Figure 2 show T cell: Renca cell conjugates?

8. Do Renca cells express ligands for any of the inhibitory receptors analysed?

9. Figure 8. GFP-CI-MRPcyt – this construct is not described well

Comments on the Quality of English Language

The manuscript would benefit from some language editing.

Author Response

see attached file please

Reviewer 3 Report

Comments and Suggestions for Authors

The authors used various cell imaging techniques to analyze the static and dynamic distribution characteristics of the five inhibitory receptors within T cells and on the plasma membrane of T cells. The results are credible and meaningful, providing intuitive clues to further explore the different roles of these five inhibitory receptors in T cell activation and effects. Several points that are worth considering are that:

1.The intracellular distribution characteristics of the five inhibitory receptors in naive T cells appear similar to that in the activated T cells stimulated by antibodies.

2. In Figure 3 and Figure 4, the tables displaying the statistical difference between groups are not easy to visually understand, can it be improved?

3. In Figure 4, which data belong to the T cells stimulated  by anti-TCR with or without extracellular domain of ICAM?

Author Response

see attached file please

Reviewer 4 Report

Comments and Suggestions for Authors

In this article entitled "Five inhibitory receptors display distinct vesicular distributions in T cells", the authors clearly and beautifully demonstrated the subcellular localization of 5 major inhibitory receptors within T cells. Using well established quantitative imaging, the authors demonstrated their hypothesis about the cellular distributions of studied molecules, even though the mechanical insights are not shown. It is strongly recommended that this article be published in its current form, as it provides valuable insights into the field of cancer immunotherapy. I have a minor suggestion that it would be helpful if the authors used better images or another way of representing Figure 4B. 

Author Response

see attached file please

Reviewer 5 Report

Comments and Suggestions for Authors

In this study, Lu and colleagues explore the sub cellular localization of 5 major inhibitory receptors on activated CD4 and CD8 T cells.  The results, largely gained through image analysis using some very nice methodology at the cutting edge, suggests that the location of these molecules is distinct, which may have implications for regulation of surface expression, etc.  Given the importance of these molecules as targets for therapeutics, this work is significant.  The experiments appear to be well controlled and the data is convincing, though a high level of expertise is required to fully understand the imaging techniques and analysis.

What detracts from the significance is the very artificial in vitro systems used to activate the cells, combined with activation steps required for some of the techniques (transduction, for example).  The translational relevance of the patterns identified may thus not be directly applicable to all or any disease states.  The authors themselves hint at this, for example when discussing results seen for PD-1 and LAG-3 and speculating that this could be caused by continuous TcR stimulation used to activate the cells.  There are alternative strategies for generating effector cells without continuous TcR stimulation over the course of several days (for example in cultures using purified T cells and irradiated APC that only persist in culture for 2 days or so instead of throughout).  The authors should discuss these possibilities more clearly in the paper.  

Furthermore, while it might also be viewed as a positive, the lack of real kinetic analysis of expression (relative to the state of T cell activation) can be viewed as a weakness.  This is particularly relevant given the different times during which different checkpoint inhibitors are thought to act.  The best example may be that CTLA-4 is thought more to act during initial priming events while regulation by PD-1 is thought to gain importance when highly activated T cells leave the priming environment and encounter antigen in peripheral tissues several days later.  Though it seems a point that might be addressed experimentally in future studies, the authors should discuss this point in their discussion.

Finally, a limitation of the study is its analysis of murine vs human T cells.  Whether the results from mouse to human are directly translatable is, as always, an open question.  I recommend the title be amended to read "..... distributions in murine T cells" to make this point clear.

In summary, there are many interesting points raised in this manuscript that might have impact on improving therapeutic regimes in which checkpoint inhibitors are targeted to improve clinical outcomes.  Additional experiments to address potential complicating issues such as kinetics and duration of T cell receptor signaling would add to the translational potential of the studies, but this reviewer feels that the data are sufficient as is to provide the field with an interesting insight.  Addressing the concerns that have been raised in the discussion and/or other parts of the manuscript will be sufficient.

Author Response

see attached file please
